# HIERARCHICAL GFLOWNET FOR CRYSTAL STRUCTURE GENERATION

## ABSTRACT

Discovering new solid-state materials necessitates the ability to rapidly explore the vast space of crystal structures and locate stable regions. Generating stable materials with desired properties and composition is a challenging task because of (a) the exponentially large number of possibilities when the elements from the periodic table are considered along with vast variations in their 3D arrangement and corresponding lattice parameters and (b) the rarity of the stable structures. Furthermore, materials discovery requires not only optimized solution structures but also diversity in the configuration of generated material structures. Existing methods have difficulty when exploring large material spaces and generating significantly diverse samples with desired properties and requirements. We propose Crystal Hierarchical Generative Flow Network (CHGlownet), a new generative model that employs a hierarchical exploration strategy with Generative Flow Network to efficiently explore the material space while generating the crystal structure with desired properties. Our model decomposes the large material space into a hierarchy of subspaces of space groups, lattice parameters, and atoms. We significantly outperform the iterative generative methods such as Generative Flow Network (GFlowNet) and Physics Guided Crystal Generative Model (PGCGM) in crystal structure generative tasks in validity, diversity, and generating stable structures with optimized properties and requirements.

## 1 INTRODUCTION

Discovering new solid-state materials plays a central role in advancing various technologies, including energy generation and storage, and semiconductor electronics (Berger, 2020; Noh et al., 2019). Each unique crystal structure exhibits properties useful for specific applications. For example, superconductive perovskite structure is used in circuit board elements for computers. Generating material structures that meet given property requirements poses a set of unique challenges. The key is to generate crystal structures with the repeating arrangement of atoms in three-dimensional space throughout the material. A crystal structure is determined by how atoms are arranged within the unit cell specified by its lengths and angles. However, the inter-atom interactions are not confined within the unit cell but also with adjacent unit cells. These characteristics make the search space of crystal structures significantly larger and more complex compared to well-studied molecular search space. The number of known crystal structures, both experimental and hypothetical, is around 3 million curated from AFlow (Mehl et al., 2017; Hicks et al., 2019; 2021) and Material Project (Jain et al., 2013), which is tiny compared to billions of molecules from the Znic dataset (Irwin & Shoichet, 2005). The limited data undermines modern data-driven methods to learn the crystal structure representation (Chithrananda et al., 2020; Liu et al., 2019), thus making crystal structure generation significantly harder than molecule generation.

We address the complexity associated with the large search space by proposing a new generative model termed Hierarchical Generative Flow Networks (HGFlowNets). HGFlowNets explores the vast search space in an efficient way. The key insight to solving the large state space problem is breaking space exploration into more meaningful hierarchical sub-tasks. Here the higher-level tasks explore more actions that are closely related to the reward function while lower-level tasks handle the configuration adjustment corresponding to the action taken at higher-level tasks. Since the exploration starts at the highly general concept level, it can learn a more meaningful policy that corresponds to the target reward function. With more meaningful actions taken at a higher level,

the policy networks can focus on searching the actions in a significantly smaller sub-space that corresponds to actions of high-level tasks instead of exploring the whole space.

Three key concepts can help the generative models to search in the material space effectively. Firstly, the crystal structure class imposes a set of symmetry operations and geometrical characteristics on the atom position and lattice parameters, thus effectively reducing the material search space. In addition, each crystal structure class is associated with relevant properties, for example, perovskite structure with conductivity. Secondly, searching for stable structures on the non-smooth energy landscape defined by quantum mechanics requires the generative models to explore and generate diverse sample sets to avoid getting stuck in one single mode and local minima. Thirdly, we apply a bond constraint on the atom pairs in the generated crystals whereby atoms should not be closer to each other than specific bond thresholds. These thresholds are obtained by calculating the minimum bond distance for every atom pair in all the materials in the MaterialsProject database. Given the diversity of materials in the database, it is reasonable to assume that minimum distances obtained from this database are threshold distances for the ground state structure of the respective atoms in any new crystal structure.

More specifically, we model the whole material space in a hierarchical structure. The highest level is the crystal structure class. Since the crystal structure class is related to relevant properties of the material, exploring the crystal structure class can efficiently lead to more optimal properties of the generated structure. In this case, we specifically use the space group of the crystal structure to effectively reduce the 3D atom space exploration to a more meaningful high-level exploration and generate a high-symmetry crystal structure. The next level searches the unit cell lattice parameters and atoms' configuration given the space group. The space group of the crystal structure also imposes constraints on the lattice parameters and atoms' position, thus reducing the lattice parameters' and atoms' search space. Choosing the positions one by one creates a long horizon trajectory, making it difficult to learn a policy that generates high symmetry structures that match proportionally to the reward function. The space group's symmetry operation can immediately replicate the atoms over the unit cell, thus reducing the trajectory length and making it easier for the policy network to learn how to generate a high-symmetry crystal structure.

In this work, we apply our proposed Hierarchical Generative Flow Network to the crystal structure generation task and refer to the resulting model as **Crystal Hierarchical Generative Flow Network** (CHGFlowNet). Our main contributions are:

- We propose a generative model that can search effectively in a large search space by modeling the state space in a hierarchical structure.
- We incorporate the physical knowledge curated from the large material databases into our generative model to generate more stable structures.
- We validate the hierarchical structure state space and physics priors of our proposed generative model in the crystal structure generation task to show the efficiency in material space exploration as well as the stability of the generated structures.

## 2 RELATED WORKS

Crystal structure generation frameworks can be classified into three main approaches based on the initial step and their search strategy:

**Element substitution** Given the templates from the ICSD (Belkly et al., 2002), the elements within the crystal structure are substituted with another element type having similar properties before being optimized by density functional theory (DFT) (Hautier et al., 2011; Wang et al., 2021; Wei et al., 2022). Generally, this approach is computationally expensive and relies on domain knowledge in element substitution.

**Material distribution sampling** Deep generative models learn from the distribution of stable crystal structures derived from experimentally known ones and sample from the distribution new crystal structures. An important model class is based on Variational AutoEncoders (VAEs) (Xie et al., 2022; Court et al., 2020; Noh et al., 2019) which learns and samples from the stable material distribution. Another model class derives from Generative Adversarial Networsk (GANs) (Zhao et al., 2021; Kim et al., 2020) which use a generator to create hypothetical crystal structures based on composition

or space group symmetry, and a discriminator to differentiate those generated structures from the real samples. Diffusion models coupled with symmetry-aware probabilistic model (Luo et al.) or periodic-E(3)-equivariant denoising model (Jiao et al.) target the invariance of crystal structure. This data-driven approach has difficulty in generating high symmetry structures and in out-of-distribution generalization. Also, the generation process is solely based on learning from the known distribution, leaving minimal room for domain knowledge and human intervention during the generation process.

**Iterative generation** Crystal structures can be decomposed into compositional objects, and constructed step by step using reinforcement learning (RL). A recent work (Zamaraeva et al., 2023) only applies RL for crystal structure prediction and optimization. Our work is the first to apply the RL-based technique to explore the entire material space. The key advantage of this RL-based approach is its high flexibility, which allows for the incorporation of domain knowledge into the action and state space and shaping the reward function.

## 3 PRELIMINARIES

### 3.1 CRYSTALLOGRAPHIC SPACE GROUP

The crystal structure is the repeating arrangement of the atoms within a unit cell in 3D. Formally, the unit cell of $N$ atoms is a triplet $(L, A, X)$ of lattice parameters $L$, atom list $A$, and atom coordinates $X$. There are 6 lattice parameters $L = (a, b, c, \alpha, \beta, \gamma) \in \mathbb{R}^6$ describing 3 lengths and 3 angles of the unit cell, respectively. The atoms list $A = (a_1, ..., a_N)$ describes the elements. The atoms' coordinates $X \in \mathbb{R}^{N \times 3}$ describe the positions of the atoms within the unit cell, which can be Cartesian or fractional.

The space group of crystal structure consists of a list of symmetry transformations to the atoms within the unit cell. In crystallography, there are 230 space groups (Glazer et al., 2012). The symmetry level of the crystal structure also increases with the space group number: Group 1 has the lowest symmetry meaning all atom's positions and lattice parameters are free without following any symmetry operations and constraints.

Each space group has geometrical characteristics defined in lattice angles and lengths that can be used as constraints to limit the parameters search space. The list of geometrical characteristics is provided in the Supplement Table 5. Given the space group $G_s$, the elements of the space group $g \in G_s$ are a set of symmetry operations. A crystallographic orbit of an atom $o = (x_o, a_o)$ with coordinate $x_o$ and element $a_o$ is defined as

$$O_{G_s}(x_o) = \{g \cdot x_o \mid g \in G_s\} \tag{1}$$

where $g \cdot x_o$ denotes the application of the symmetry operation $g$ on the atom $o$ within the unit cell. From a reference atom $o$, we can obtain a set $O_{G_s}$ of equivalent points.

### 3.2 GENERATIVE FLOW NETWORK

The Generative Flow Network (GFlowNet) is a generative model designed to sample candidates proportional to their target rewards. The framework has been successfully applied to many fields such as molecular discovery (Bengio et al., 2021), protein sequence discovery (Jain et al., 2022), causality (Deleu et al., 2022), and continuous control (Li et al., 2023). GFlowNet models the sampling process of the compositional object $s$ as the directed acrylic graph (DAG) $G = (S, A)$ where $S$ is the set of states and $A$ is the state transition which is the subset of $\mathcal{S} \times \mathcal{S}$. The sampling process starts with the initial state vertex $s_0 \in S$ with no incoming edge. and stops at the sink state vertex $s_n \in S$, $n$ is the sampling trajectory length, with no outgoing edge. GFlowNet learns a policy function $\pi$ that can sample the object $x$ with the probability proportional to the non-negative reward function. GFlowNets constructs the object step by step, from the initial state $s_0$ to the sink state $s_n$, forming a trajectory $\tau = s_0, .., s_n, \tau \in \mathcal{T}$ where $\mathcal{T}$ is the trajectory set. For any state $s$ in the trajectory, we can define the flow of the state as $F(s) = \sum_{\tau \ni s} F(\tau)$ and the flow of the edge $s \to s'$ as $F(s) = \sum_{\tau \ni s \to s'} F(\tau)$ (Bengio et al., 2021). The forward policy that maps the probability of transition from the current state $s$ to the next state $s'$ is given as $P_F(s'|s) = \frac{F(s \to s')}{F(s)}$. The backward policy mapping probability transition to the previous state $s$ given the current state $s'$ is $P_B(s|s') = \frac{F(s \to s')}{F(s')}$. The training objective of the GFlowNets is flow matching consistency where

the incoming flow is equal to the outgoing flow, $\sum_{s'' \to s} F(s'' \to s) = F(s) = \sum_{s \to s'} F(s \to s')$, for all states. Malkin et al. (2022) propose trajectory balance to deal with the long trajectory credit assignment problem. Given the trajectory $\tau = (s_0 \to s_1 \to ... \to s_n)$, the forward probability of a trajectory is defined as $\prod_{t=1}^{n} P(s'|s)$. The trajectory balance constraint is defined as:

$$Z \prod_{t=1}^{n} P_F(s_t|s_{t-1}) = F(x) \prod_{t=1}^{n} P_B(s_{t-1}|s_t) \tag{2}$$

where $P(s_n = x) = \frac{F(x)}{Z}$. Then the trajectory balance objective is defined as:

$$\mathcal{L}_{TB}(\tau) = \left( \log \frac{Z_\theta \prod_{t=1}^{n} P_F(s_t|s_{t-1}; \theta)}{R(x) \prod_{t=1}^{n} P_B(s_{t-1}|s_t; \theta)} \right)^2 \tag{3}$$

## 4 PROPOSED METHOD

We now describe our CHGFlowNets, a hierarchical generalization of GFlowNets to operate on the hierarchical structure state space of crystals. In the hierarchy, lower states represent discrete concepts constrained by the higher states that represent more abstract concepts. The key to the design of CHGFlowNets is to allow efficient exploration of high-symmetry crystal structures with desired properties.

In particular, the symmetry is defined on the space group structure outlined in Sec. 3.1 that imposes constraints on the lattice parameters. In the space group, for $n$ symmetry operations, one can identify the positions and elements of other $n-1$ atoms given only the (coordinates, element) pair of one atom $o = (x_o, a_o)$. This effectively reduces the number of searches $n$ times. The lattice parameters shape the unit cells containing the atoms, thus affecting the atoms' distance and interaction. Therefore, we place the atom state below the lattice parameters state in our hierarchical state structure. Fig. 1 describes the overall structural design.

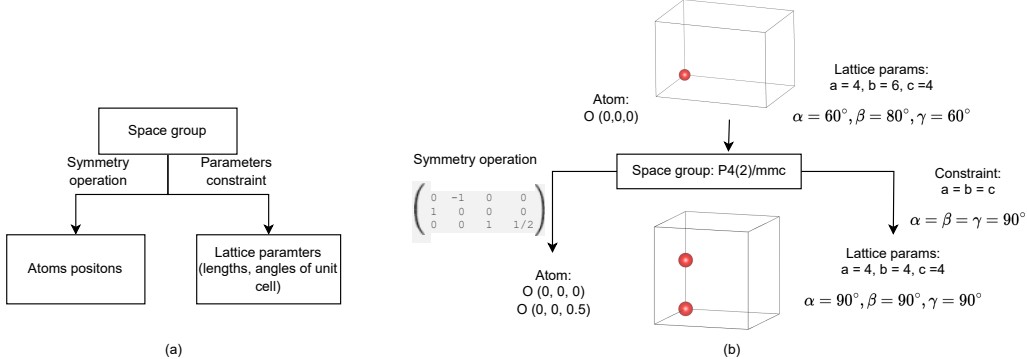

Figure 1: (a) Hierarchical crystal structure state. The space group level provides the set of symmetry operations for atoms' positions and lattice parameters constraints. (b) An example of applying the hierarchical state space. The current state has one Oxygen atom at position (0, 0, 0), lattice parameters $a = 4, b = 6, c = 4, \alpha = 60°, \beta = 80°, \gamma = 60°$, and $P1$ spacegroup. The action of choosing space group P4(2)/mmc provides a symmetry operation to generate another Oxygen atom at position (0, 0, 0.5). The lattice parameter constraints reduce the unit cell's length search space from $\mathbb{R}^3$ to $\mathbb{R}$ and make the unit cell's angles constant at $90°$

### 4.1 STATE ENCODING.

We represent the crystal unit cell $(L, A, X)$ described in Sec. 3.1 as a directed graph $\mathcal{G} = (\mathcal{V}, \mathcal{E})$ of node feature matrix $\mathcal{V}$ and edge matrix $\mathcal{E}$. The node features include the atom's atomic number and its fractional coordinates within the unit cell. The edges are determined by k-nearest neighbor with the maximum number of neighbors being 12 and the radius cut-off is $8.0\mathring{A}$. The graph representation

is learned using the Graph Convolution Networks (GCNs) (Kipf & Welling, 2017), equipped with skip connections to allow deep layers:

$$H^l = \sigma(H^{l-1} + F^l(H^{l-1})), \quad \text{where} \tag{4}$$

$$F^l(H^{l-1}) = W^{l-1}\sigma(\text{GCN}(H^{l-1}, \mathcal{E})), \tag{5}$$

where $W$ is the learnable weight matrix, $l$ is the layer's index, and $\sigma$ is a non-linear ReLU activation function. The GCN readout function uses max pooling, followed by a two-layer MLP to output graph-level representation $h_\mathcal{G}$.

Lattice parameters are encoded using a multi-layer perceptron as:

$$h_\mathcal{L} = \text{MLP}\left([l_1, l_2, l_3, \sin(\alpha), \cos(\alpha), \sin(\beta), \cos(\beta), \sin(\gamma), \cos(\gamma)]\right) \tag{6}$$

where $l_1$, $l_2$, $l_3$ are the lattice lengths, and $\alpha, \beta, \gamma$ are the angle of the lattice angle. Finally, the crystal structure state is simply $s_M = [h_\mathcal{G}; h_\mathcal{L}]$.

## 4.2 HIERARCHICAL POLICY

Our hierarchical policy consists of two levels: The high-level decision-making policy operating on the space groups, and the low-level execution policy operating on the atom-lattices (see Fig. 2). The space group policy chooses the space group and applies corresponding constraints on the atom-lattice policy actions. The corresponding hierarchical state space is decomposed as $s = (s_{sg}, s_{al})$, where $s_{sg}$ is the space group state and $s_{al}$ is the atom-lattice state. The latter consists of lattice parameters $s_{lp}$, atoms' coordinate $s_{ac}$, and atoms' type $s_{at}$ states.

Then we define the probability of transitions as:

$$P(s'|s) = P(s'_{sg}, s'_{al}|s_{sg}, s_{al}) \tag{7}$$

$$P(s'_{sg}, s'_{al}|s_{sg}, s_{al}) = P(s'_{al}|s_{sg}, s_{al}, s'_{sg})P(s'_{sg}|s_{sg}, s_{al}) \tag{8}$$

Then given the trajectory $\tau = (s_0 \rightarrow s_1 \rightarrow ... \rightarrow s_n)$, we then have the trajectory balance constraint (Eq. 2) with the decomposed state as:

$$Z\prod_{t=1}^n P_F(s^t_{al}|s^{t-1}_{sg}, s^{t-1}_{al}, s^t_{sg})P_F(s^t_{sg}|s^{t-1}_{sg}, s^{t-1}_{al}) \tag{9}$$

$$= F(x)\prod_{t=1}^n P_B(s^{t-1}_{al}|s^t_{sg}, s^t_{al}, s^{t-1}_{sg})P_B(s^{t-1}_{sg}|s^t_{sg}, s^t_{al}) \tag{10}$$

The space group forward and backward transition probabilities $P_F(s^t_{sg}|s^{t-1})$ and $P_B(s^{t-1}_{sg}|s^t)$ are parameterized by the multinoulli distribution defined by the logits output of the space group policy networks. The lattice parameters state $s_{lp}$ transition probability is parameterized by the Gaussian distribution defined by the mean $\mu$ and variance $\sigma^2$. The atom fraction coordinates state $s_{ac}$ transition probability is parameterized by the Multivariate Gaussian distribution given by the mean $\mu$ and covariance matrix $\Sigma$. The atom type state $s_{at}$ transition probability is parameterized by the multinoulli distribution. These transition probabilities are used in the trajectory sampling process A.3.2.

The model is trained using the trajectory balance training objective in Eq. 3.

## 4.3 PHYSIC-INFORMED REWARD FUNCTION

At the terminal state of the trajectory, a reward is returned by a non-negative function $R(x)$, providing feedback on the generated crystal structure, especially its validity and stability. The reward function is composed of the following terms:

The **formation energy term** dictates that a stable crystal structure will have a negative formation energy, thus defined as $R_e(x) = e^{-E(x)}$, where $E(x)$ is the predicted formation energy per atom given by the prediction model A.3.6.

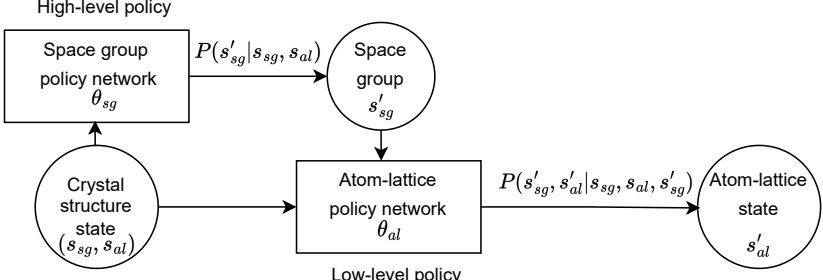

Figure 2: Hierarchical policy for crystal structure state. First, the crystal structure graph state $s$ is decomposed into space group state $s_{sg}$ and atom-lattice state $s_{al}$. Then the transition probability $P(s'_{sg}|s_{sg}, s_{al})$ in Eq. 8 is given by the space group policy network $\theta_{sg}$. The transition probability $P(s'_{al}|s_{sg}, s_{al}, s'_{sg})$ in Eq. 8 is given by the atom-lattice policy network $\theta_{al}$.

The **bond distance preferences term** $R_b$ is defined by the distance between any atom pairs $(a_i, a_j)$. The term consists of two validation conditions which are the minimum distance constraint and neighbour distance constraints. The minimum distance constraint term is defined as

$$R_{min}(x) = \begin{cases} \alpha_p, & \text{if } \exists i, j \in x : d(a_i, a_j) < d_{min}(a_i, a_i) \\ 1, & \text{otherwise.} \end{cases} \tag{11}$$

where $d(a_i, a_j)$ is the distance between two $a_i$ and $a_j$ atoms, $d_{min}(a_i, a_i)$ is the minimum distance retrieved from the database, and $\alpha_p < 1$ is the penalty term hyper-parameter.

The neighbor distance term is defined as:

$$R_{max}(x) = \begin{cases} \beta_p, & \text{if } \exists j \in x, \forall \text{nei}(j), d(a_j, \text{nei}(a_i)) > d_{max}(a_j, \text{nei}(a_i)) \\ 1, & \text{otherwise.} \end{cases} \tag{12}$$

where $d(a_i, a_j)$ is the distance between two $a_i$ and $a_j$ atoms, $\text{nei}(a)$ is the neighbours atoms, $d_{max}(a_i, a_i)$ is the minimum distance retrieved from the database, and $\beta_p < 1$ is the penalty term hyper-parameter.

The bond distance preference is defined as:

$$R_{bond}(x) = \begin{cases} \gamma_p, & \text{if } R_{max}(x) < 1 \text{ and } R_{min}(x) < 1 \\ \alpha_p & \text{if } R_{min}(x) < 1 \\ \beta_p & \text{if } R_{max}(x) < 1 \\ 1, & \text{otherwise.} \end{cases} \tag{13}$$

The **density term** is defined upon the structure density. Because the bond distance preference term applies a strict penalty for structures violating the minimum distance constraint, the generative model tends to generate a structure with long distances between atoms pairs with only a few neighbor atoms. This leads the model to generate gas with low density rather than solid-state density. On the other hand, structures with very high density (i.e. larger than 10) are unlikely to be realistic as the units are crowded with atoms causing a very high formation energy. We measure the generated structure density $P(x)$ and define the density term as the Gaussian function of structure density:

$$R_P(x) = ae^{\frac{-(P(x)-b)^2}{2c^2}} \tag{14}$$

The **composition validity term** is defined as $R_{comp}(x) = 1$ for valid composition and $R_{comp}(x) = 0$ otherwise. Finally, the **physic-informed reward function** is composed as:

$$R(x) = (R_e(x) + R_P(x)) * R_{bond}(x) + R_{comp}(x) \tag{15}$$

## 5 EXPERIMENTS

**Battery material discovery task** Motivated by the search for light-weight, transition-metal free cation battery materials, we explore the space of possible materials that can be made from the light elements Be, B, C, N, O, Si, P, S and Cl, and one of the three alkali metals Li, Na and K. This space of materials constitutes materials that can be utilised in lithium-ion, sodium-ion, and potassium-ion battery materials, respectively. Of particular interest to our work is the generation of new transition-metal free solid-state electrolyte materials that can be incorporated in solid-state lithium, sodium, and potassium batteries.

**Baselines** We compare our CHGFlowNet with the latest crystal generation model of PGCGM (Zhao et al., 2023) and the Generative Flow Networks (GFlowNets), which is a flat version of our method. The PGCGM is a GAN-based method that uses a physics-informed loss function defined by the distance between atoms.

As the original GFlowNets only work on the discrete space, we follow the recent work on continuous GFlownet (Lahlou et al., 2023) to work on the continuous space of the atoms' coordinates and lattice parameters. The model has a single-level policy network that outputs space group, lattice parameters, atoms' coordinates, and atoms' type. Note that this is the first time GFlowNet has been applied successfully for crystal generation.

### 5.1 MATERIAL VALIDITY

We evaluate the proposed method and baseline methods on the validity of the generated crystal structure, measured based on three criteria. We follow the previous work (Zhao et al., 2023) for validity measurements a) *CIFs validity* is the percentage of Crystallographic Information Files (CIFs) of generated crystal structure that is readable by pymatgen (Ong et al., 2013) b) *structure validity*: a structure is valid as long as the minimum distance between any two atoms is more than $0.5\mathring{A}$ c) *composition validity*: a composition is valid if the overall charge computed by SMACT Davies et al. (2019) is neutral. As seen in Tab. 1, CIFs validity is an easy condition, and all three methods can achieve a validation rate close to one. Both GFlowNet and CHGFlowNet structure and composition validities are close to one, highlighting the effectiveness of learning reward-based exploration. The structure and composition validities are used in the reward function described in Sec. 4.3. In contrast, PGCGM faces the common problem of sampling from data-induced distribution without any refinement, which is low structure validity.

Table 1: Validity of the generated structures. We evaluate the top 1000 crystal structures ranked by the reward function after $10^5$ states visited.

| Method | Validity | | |
|---|---|---|---|
| | CIF ↑ | Structure ↑ | Composition ↑ |
| PGCGM | 1 | 0.101 | 0.747 |
| GFlowNet | 1 | 0.995 | 0.998 |
| CHGFlowNet | 1 | 0.998 | 0.986 |

### 5.2 MATERIAL DIVERSITY AND FORMATION ENERGY

Table 2: Diversity and average of formation energy of the generated structures. We evaluate the top 1000 crystal structures ranked by the reward function after $10^5$ states visited.

| Method | Diversity | | | Formation Energy ↓ | % stable ↑ |
|---|---|---|---|---|---|
| | Crystal family ↑ | Composition ↑ | Structure ↑ | | |
| PGCGM | 2.467 | 2141.196 | 0.524 | 4.558 | 62.5 |
| GFlowNet | 2.544 | 3205.639 | 0.651 | 1.433 | 73.4 |
| CHGFlowNet | 2.616 | 3037.808 | 0.761 | 0.882 | 89.9 |

**Diversity** Following the previous works (Xie et al., 2022; Zhao et al., 2023), we evaluate both the structure and composition diversity of the generated crystal structures. The *structure diversity* is defined as the average pairwise Euclidean distance between the structure fingerprint of any two generated materials (Pan et al., 2021). The *composition diversity* is defined as the average pairwise distance between the composition fingerprints of any two generated materials (Pan et al., 2021). More details are provided in A.3.4. We further use the crystal family defined in Supplement Tab. 5 which is a group of space groups sharing some special geometric characteristics. The *crystal family diversity* is defined as the Shannon–Wiener index (Shannon, 1948) of the number of generated structures in each crystal family.

We evaluate the diversity, average formation energy structures, and the percentage of stable structures discovered by the models, and report the results in Tab. 2. A stable structure is defined as a structure with formation energy per atom smaller than 2eV/atom. All three methods can find stable structures more than 60% of the time, but the GFlowNet family has more hit rate of 73.4% and 89.9%. Since PGCGM learns a crystal distribution from data and samples from it, the diversity of generated structures is low. And without further optimisation step, its formation energy is rather high on average. Compared with the flat GFlowNets, our CHGFlowNets find crystals with lower formation energy, and more diversity in terms of crystal family and structure.

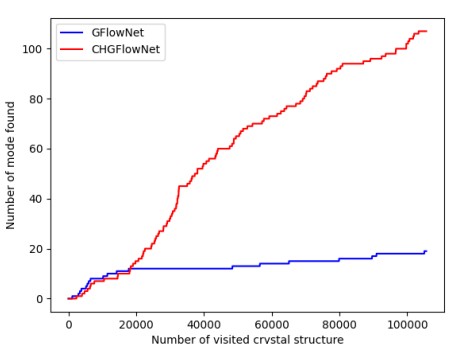

Figure 3: Comparison of CHGFlowNet and GFlowNet in exploring crystal modes using 3 steps. A mode is defined as a valid crystal structure with negative formation energy. A step is an action of choosing one atom in the spacegroup-lattice-atom hierarchical state space.

**Material mode exploration** To evaluate the speed of exploring the material space and finding the valid material structures, we count the number of modes found by the generated crystal structures plotted against the number of states visited in Fig. 3. We define a unique *mode* as a valid crystal structure satisfying conditions A.3.5. The results show that using our hierarchical model improves the speed of mode discovery compared to the flat variant.

## 5.3 STABILITY OF GENERATED MATERIALS

Table 3: Match rates of the generated crystal paired with structures optimized by M3GNet (Chen & Ong, 2022)

| Methods | Match rate ↑ |
|---|---|
| PGCGM | 0.625 |
| GFlowNet | 0.678 |
| CHGFlowNet | 0.753 |

It is a common practice to relax the generated structures to seek the lower the potential energy surface using DFT calculation iteratively. As DFT calculation is expensive, it is desirable to generate a structure that is close to energy minima. In this experiment, we compare the generated crystal structure with its optimized structure. We use the M3GNet framework (Chen & Ong, 2022) to iteratively optimize the energy predicted by the potential surface energy model. Examples of generated structures and their corresponding optimized structures are shown in Fig. 4.

**Match rate** We follow the previous work in (Zhao et al., 2023) to evaluate the match rate of crystal structure relaxation. A structure $m$ and its optimized structure $m'$ are matched if their atoms' translation and angle are within tolerance thresholds, indicating that the generated structure is close to the optimal, thus is more stable. We use the matching algorithm provided by pymatgen library (Ong et al., 2013) in StructureMacher with $10^o$ angle tolerance, 1.0 fractional length tolerance, and 1.0 site tolerance. The match rate is the fraction of the number of matched structures on the total number of generated structures.

The results reported in Tab. 3 show that our proposed method can produce more structures (at the rate of 75.3%) that are nearly optimal in terms of total energy compared to GFlowNets and PGCGM.

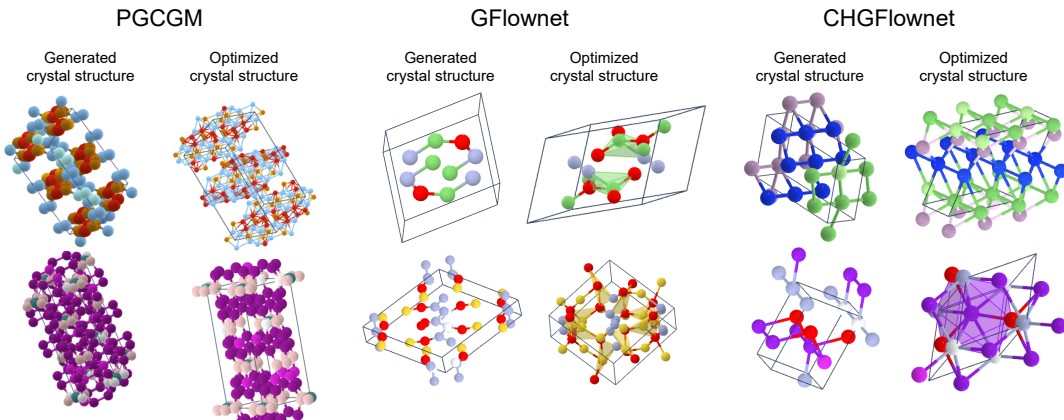

Figure 4: Examples of generated crystal structures and the corresponding structure optimized by M3GNet framework (Chen & Ong, 2022).

## 5.4 ABLATION STUDY

To demonstrate the ability to guide the generative model to generate more stable structures, we perform the ablation study on the reward function's terms of Eq. 15. The generated crystal structures are relaxed using M3GNet optimization framework (Chen & Ong, 2022).

Table 4: The ablation study on the impact of reward function terms on the crystal structure stability. Match rates of the generated crystal paired with structures optimized by M3GNet (Chen & Ong, 2022)

| Methods | Match rate ↑ |
|---|---|
| All terms | 0.753 |
| W/o density | 0.617 |
| W/o bond score | 0.739 |
| W/o formation energy | 0.734 |

The results reported in Tab. 4) show that all the density term, bond score term, and formation energy term are necessary for the model to generate more stable structures. During the relaxation process, both atoms' positions and lattice parameters are adjusted to lower the total energy and the force of the crystal structure. As a result, the density and the distance between atoms are changed significantly. By using the preference density and distance distilled from prior knowledge such as the relaxed crystal structure dataset, the generation model is able to place atoms and adjust the lattice parameters to maintain the preference distances between atoms.

## 6 CONCLUSION

We have proposed CHGFlowNet, a Hierarchical Generative Flow Network for crystal structure generation, aiming at rapid exploration of the exponential crystal space and simultaneously satisfying physics constraints. CHGFlowNet is built on a hierarchical state space, allowing for multi-level policy networks to operate on action abstraction. It effectively exploits the high-symmetry in the crystal structure space, defining space transformation groups. The framework is rather flexible, allowing domain experts to embed physics and chemistry knowledge to guide the generation process through space structure design and reward engineering. CHGFlowNet demonstrates its superiority in efficiency in exploration, diversity, and stability in the generated crystal structures. While our focus is on materials discovery, our hierarchical state space and policy can easily extend to other tasks and deeper hierarchy such as continuous control and robotics where multiple layers of abstraction and discreet states and actions are required.

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

# A APPENDIX

## A.1 GEOMETRICAL CHARACTERISTICS

Table 5: Geometrical characteristics of space groups in terms of lattice angles or lengths

| Space group | Crystal family | Lengths constraints | Angles constraints | Parameters search space |
|---|---|---|---|---|
| 1-2 | Triclinic c | None | None | $a, b, c, \alpha, \beta, \gamma$ |
| 3-15 | Monoclinic | None | $\alpha = \beta = 90°$ | $a, b, c, \gamma$ |
| 16-74 | Orthorhombic | None | $\alpha = \beta = \gamma = 90°$ | a, b, c |
| 75-142 | Tetragonal | $a = b$ | $\alpha = \beta = \gamma = 90°$ | a, c |
| 143-194 | Hexagonal | $a = b$ | $\alpha = \beta = 90°, \gamma = 120°$ | a, c |
| 195-230 | Cubic | $a = b = c$ | $\alpha = \beta = \gamma = 90°$ | a |

## A.2 CONTINOUS GFLOWNET ASSUMPTIONS

The soundness of the theory of continuous GFlownet relies on a set of assumptions:
- The structure of the state space must allow all states to be reachable from the source state $s_0$.
- The structure must ensure that the number of steps required to reach any state from $s_0$ is bounded.
- The learned probability measures need to be expressed through densities over states, rather than over actions.

Our crystal structure state space and training framework satisfy these assumptions as:
- Given the $s_0$ as the empty crystal structure with space group 1 (lowest symmetry without any constraints on the lattice parameters and no symmetry operation) and initial lattice params, the measureable pointed graph is defined as $\mathcal{S} = s_0 \cup [min_l, max_l]^3 \cup [min_a, max_a]^3 \cup [0, 1]^T$ where $min_l$, $max_l$ are minimum and maximum lattice length, $min_a, max_a$ are minumum and maximum lattice angle.
- We have the upper limit $T$ for the trajectory length during the trajectory sampling process. Therefore the number of step to reach any states is bounded by $T$.
- The forward $p_F$ and backward $p_B$ policy networks learn the distribution over states $P_F(s_{al}^t | s_{sg}^{t-1}, s_{al}^{t-1}, s_{sg}^t), P_F(s_{sg}^t | s_{sg}^{t-1}, s_{al}^{t-1}), P_B(s_{al}^{t-1} | s_{sg}^t, s_{al}^t, s_{sg}^{t-1}), P_B(s_{sg}^{t-1} | s_{sg}^t, s_{al}^t)$

## A.3 IMPLEMENTATION DETAILS

### A.3.1 STATE GRAPH CONSTRUCTION

We determine the edges of the crystal structure graph using k-nearest neighbor atoms within $4\mathring{A}$. The node feature is the coordinate and the atomic number of the atom.

### A.3.2 SAMPLING PROCESS

The lattice sampling process starts with the state $s_0$ which has lattice parameters as $a = b = c = 4, \alpha = \beta = \gamma = 90^o$, space group 1, empty crystal graph at state $s_0$ $\mathcal{G}_{s_0}$. The sampling process is Algorithm 1.

### A.3.3 WYCKOFF POSITIONS

We use the function Structure.from_spacegroup which, for a given space group, evaluates the Wyckoff positions in a lattice structure and gives a set of species and positions within the lattice, such that the resultant structure satisfies the symmetry operations of the space group. The Wyckoff positions are obtained from a python dictionary within pymatgen, and use the fractional coordinates of the atoms as input. The generated atoms will occupy the Wyckoff positions, but for a structure to be valid, we apply the minimum distance in the reward function to encourage the policy network to sample valid structure.

---

**Algorithm 1:** Trajectory sampling

---

**Input:** $\theta_{sg}, \theta_{al}$, T, $min_l, max_l$,
**Return:** Trajectory $\tau$, complete crystal structure $x$
$s_{al} \leftarrow \{a = b = c = 4, \alpha = \beta = \gamma = 90^o\}$;
$s_{sg} \leftarrow 1$
$\mathcal{G} \leftarrow \oslash$
Reference atom list $A_{ref} \leftarrow \oslash$
**for** *each step $t \leq T$* **do**
  Get the multinomial logits $p_{nsg} \leftarrow \theta_{sg}(s_{sg}, \mathcal{G})$.
  Sample space group $s'_{sg} \sim M_{nsg}(1, p_{nsg})$.
  Get distribution parameters
    $\mu_a, \sigma_a, \mu_b, \sigma_b, \mu_c, \sigma_c, \mu_\alpha, k_\alpha, \mu_\beta, k_\beta, \mu_\gamma, k_\gamma, \mu_f, p_{ne} \leftarrow \theta_{sg}(s'_{sg}, s_{sg}, \mathcal{G})$.
  Sample lattice parameter:
  $a \sim \mathcal{N}(\mu_a, \sigma_a^2)$
  $b \sim \mathcal{N}(\mu_b, \sigma_b^2)$
  $c \sim \mathcal{N}(\mu_c, \sigma_c^2)$
  $\alpha \sim vonMises(\mu_\alpha, k_\alpha)$
  $\beta \sim vonMises(\mu_\beta, k_\beta)$
  $\gamma \sim vonMises(\mu_\gamma, k_\gamma)$.
  Update $s'_{lp} = (a, b, c, \alpha, \beta, \gamma)$ values based constraints imposed by $s'_{sg}$.
  Sample atom fraction coordinate: $s'_{ac} \sim \mathcal{N}_3(\mu_f, \Sigma_f)$.
  Clamp sampled fraction coordinate to $[0, 1]$.
  Get the atom's element valid $mask$ based on the composition constraint.
  Set logits with mask $p_{ne}[mask] = -\infty$.
  Sample atom's element type $s'_{at} \sim M_{nsg}(1, p_{ne})$
  $A_{ref} = A_{ref} \cup (s'_{at}, s'_{ac},)$
  Add state $s'_{sg}, s'_{lp}, s'_{at}, s'_{ac}$ to trajectory $\tau$
**end**
Apply symmetry operation of space group $s_{sg}$ at the terminal state to the $A_{ref}$ to get complete
 crystal structure $x$

---

### A.3.4 DIVERSITY METRICS

Structure diversity is computed based on the CrystalNNFingerprint (CNN fingerprint) Ward et al. (2018). CrystalNNFingerprint computes the fingerprint of a given site $i$ using its coordination features and neighbors. The site $i$ neighbors are determined by the CrystalNN neighbor-finding algorithm. The fingerprint of a crystal structure is the average of fingerprints of all sites.

Composition fingerprint is computed using the statistics of Magpie, computed by element stoichiometry Ward et al. (2018; 2016). We use ElementProperty.from_preset('magpie') in Matminer as material composition fingerprint.

Then we define diversity as:

$$Diversity = \frac{1}{n} \sum_{i,j \in N_{gen}} d(f_{fp}(i), f_{fp}(j)) \tag{16}$$

where $N_{gen}$ is the set generated crystal structures, $n$ is the number of structure in set $N_{gen}$. In our experiment, $N_{gen}$ is the top 1000 crystal structures ranked by reward, $f_{fp}$ is the structure fingerprint in case of structure diversity or composition fingerprint in case of composition diversity, $d$ is the Euclidean distance.

### A.3.5 UNIQUE MODE

We define a unique *mode* if it satisfies four conditions. The first condition is three types of validity defined in Sec. 5.1. The second condition is that the crystal structure satisfies both distance constraints defined in Eq.11 and Eq.12. The third condition is that the structure must have negative formation energy. The fourth condition is that the composition cannot be the same as other modes.

### A.3.6 FORMATION ENERGY PREDICTION

We use M3GNet (Chen & Ong, 2022) to predict the formation energy. As the M3GNet is only trained on the Material Project valid crystal structure, the predictions for invalid structures may be inaccurate and have abnormally low formation energy. Therefore, we put the negative cut-off for the prediction. The cut-off is -10.0 eV/atom. Any prediction lower than the cut-off is set to 10 eV/atom.

### A.3.7 HYPER-PARAMETERS

See Tab. 6 for hyper-parameter setting.

Table 6: Hyper-parameters

| Hyper-paramters | Value |
|---|---|
| Learning rate | 0.0001 |
| Learning rate Z | 0.1 |
| Optimizer | Adam |
| Learning rate scheduler $\gamma$ | 1.0 |
| Initial logZ | 0.0 |
| Batch size | 32 |
| $\alpha_p$ (Eq. 11) | 0.1 |
| $\beta_p$ (Eq. 12) | 0.01 |
| $\gamma_p$ (Eq. 13) | 0.001 |

