# HIERARCHICAL GFLOWNET FOR CRYSTAL STRUCTURE GENERATION

## ABSTRACT

Discovering new solid-state materials necessitates the ability to rapidly explore the vast space of crystal structures and locate stable regions. Generating stable materials with desired properties and composition is a challenging task because of (a) the exponentially large number of possibilities when the elements from the periodic table are considered along with vast variations in their 3D arrangement and corresponding lattice parameters and (b) the rarity of the stable structures. Furthermore, materials discovery requires not only optimized solution structures but also diversity in the configuration of generated material structures. Existing methods have difficulty when exploring large material spaces and generating significantly diverse samples with desired properties and requirements. We propose a Crystal Hierarchical Generative Flow Network (CHGlownet) that employs a hierarchical exploration strategy with Generative Flow Network to efficiently explore the material space while generating the crystal structure with desired properties. Our model hierarchically decomposes the large material space into hierarchy subspaces of space groups, lattice parameters, and atoms. We significantly outperform the iterative generative methods such as Generative Flow Network (GFlowNet) and Proximal Policy Optimization (PPO) in crystal structure generative tasks in validity, diversity, and generating stable structures with optimized properties and requirements.

## 1 INTRODUCTION

Discovering new solid-state materials plays an important role in advancing various technologies, including energy generation and storage, and semiconductor electronics (Berger, 2020; Noh et al., 2019). Each unique crystal structure exhibits properties useful for specific applications. For example, superconductive perovskite structure is used in circuit board elements for computers. Discovering crystal structures with desired properties is crucial for material discovery. While there are many popular generative models for molecules, material generation poses a set of difficult and unique challenges. Differing from molecules formed by a small set of atoms with bonds clearly defined by their valence, materials are formed as a crystal structure with the repeating arrangement of atoms in three-dimensional space throughout the crystal. The crystal structure is determined by how atoms are arranged within the unit cell which is defined by its length and angle. In addition, the atoms' interactions are not limited to the unit cell but also with adjacent unit cells. These two differences make the search space of crystal structure significantly larger and more complex compared to molecule search space. The number of training samples of crystal structure is around 3 million experimental and hypothetical materials curated from AFlow (Mehl et al., 2017; Hicks et al., 2019; 2021), Material Project (Jain et al., 2013) compared to billions of molecules from the Znic dataset (Irwin & Shoichet, 2005). The limited data samples make it difficult to learn the crystal structure representation using conventional methods (Chithrananda et al., 2020; Liu et al., 2019), thus making the crystal structure generation significantly harder than molecule generation.

We address the complexity associated with the large search space by proposing a Hierarchical Generative Flow Network based on Generative Flow Networks (GFlowNets) and extend it to exploit structure in the search space in an efficient way. The key insight to solving the large state space problem is breaking space exploration into more meaningful hierarchical sub-tasks with the higher-level tasks exploring more meaningful actions that are closely related to the reward function while lower-level tasks handle the configuration adjustment corresponding to the action taken at higher

level. Since the exploration starts at the highly general concept level, it can learn a more meaningful policy that corresponds to the target reward function. With more meaningful actions taken at a higher level, the policy networks can focus on searching the actions in a significantly smaller sub-space that corresponds to actions of high-level tasks instead of exploring the whole space.

Three key concepts can help the generative models to search in the material space effectively. Firstly, different from molecules, crystal structure symmetry defines specific atoms' positions in the 3D space. The symmetry of the crystal structure is categorized into 230 space groups. Each space group imposes a set of constraints in atom positions and lattice parameters, thus effectively reducing the material search space. In addition, each crystal structure class is associated with relevant properties, for example, perovskite structure with conductivity. Taking advantage of these crystal classes can narrow down the search space significantly. Secondly, to generate stable materials, the generative framework has to search the energy landscape defined by quantum mechanics. However, the energy landscape is not smooth due to the complexity of periodic interactions of crystal structure. The generative models need to explore and generate diverse sample sets to avoid getting stuck in one single mode and local minima. Thirdly, we apply a bond constraint on the atom pairs in the generated crystals whereby atoms should not be closer to each other than specific bond thresholds. These thresholds are obtained by calculating the minimum bond distance for every atom pair in all the materials in the MaterialsProject database. Given the diversity of materials in the database, it is reasonable to assume that minimum distances obtained from this database are threshold distances for the ground state structure of the respective atoms in any new crystal structure.

With the hierarchical space structure insight in mind, we model the whole material space in a hierarchical structure. The highest level is the crystal structure class. Since the crystal structure class can related to relevant properties of the material, exploring the crystal structure class can efficiently lead to more optimal properties of the generated structure. In this case, we specifically use the space group of the crystal structure. There are only 230 space groups, thus effectively reducing the 3D atom space exploration to a more meaningful high-level exploration and generating a high-symmetry crystal structure. The next level searches the unit cell lattice parameters and atoms' configuration given the space group. The space group of the crystal structure also imposes constraints on the lattice parameters and atoms' position, thus reducing the lattice parameters' and atoms' search space. Choosing the positions one by one creates a long horizon trajectory, making it difficult to learn a policy that generates high symmetry structures that match proportionally to the reward function. The space group's symmetry operation can immediately replicate the atoms over the unit cell, thus reducing the trajectory length and making it easier for the policy network to learn how to generate a high-symmetry crystal structure.

In this work, we apply our proposed Hierarchical Generative Flow Network to the crystal structure generation task in the **Crystal Hierarchical Generative Flow Network (CHGlownet)** framework. Our main contributions are:

- We propose a generative model that can search effectively in a large search space by modeling the state space in a hierarchical structure.
- We incorporate the physical knowledge curated from the large material datasets into our generative model to generate more stable structures.
- We validate the hierarchical structure state space and physics priors of our proposed generative model in the crystal structure generation task to show the efficiency in material space exploration as well as the stability of the generated structures.

## 2 RELATED WORKS

Crystal structure generation is a challenging task due to the complex quantum mechanics and periodic interactions. Crystal structure generation frameworks can be classified into three main approaches based on the initial step and their search strategy

**Element substitution** Given the templates from the ICSD (Belkly et al., 2002), the elements within the crystal structure are substituted with another element type having similar properties before being optimized by density functional theory (DFT) (Hautier et al., 2011; Wang et al., 2021; Wei et al., 2022). Generally, this approach is computationally expensive and relies on domain knowledge in element substitution.

**Material distribution sampling** Deep generative models learn from the distribution of stable crystal structures derived from experimentally known ones and sample to generate new crystal structures. Variational autoencoder (VAE) (Xie et al., 2022; Court et al., 2020; Noh et al., 2019) learns and samples from the stable material distribution. Generative Adversarial Network (GAN) (Zhao et al., 2021; Kim et al.) use the generator to create hypothetical fake crystal structures based on composition or space group symmetry, while the discriminator is trained to differentiate real structures from those generated by the generator. This approach has difficulty in generating high symmetry structure and generalization. The generation process is solely based on learning from the distribution, leaving minimal room for domain knowledge and human intervention during the generation process.

**Iterative generation** Crystal structures can be decomposed into compositional objects, and constructed step by step using reinforcement learning (RL). Previous work (Zamaraeva et al., 2023) only applies RL for crystal structure prediction/optimization. Our work is the first time applying the RL-based technique to explore the material space. The key advantage of this approach is its high flexibility, which allows for the incorporation of domain knowledge by modifying the action and state space and shaping the reward function

## 3 PRELIMINARIES

### 3.1 CRYSTAL STRUCTURE REPRESENTATION

**Material crystal structure** The crystal structure is the repeating arrangement of the atoms in the 3D space. The crystal structure can be represented by the arrangement of the atoms within the unit cell. The unit cell is the smallest unit that repeats across 3D space, creating an infinite crystal structure. The unit cells of material $M$ having $N$ atoms can be described by the lattice parameters and its atoms. Formally, the material $M$ can be denoted $M = (L, A, X)$. The lattice parameters $L = (a, b, c, \alpha, \beta, \gamma) \in R$ are six lattice parameters describing three lengths and three angles of the unit cell. The atoms list $A = (a_1, ..., a_N) \in A^N$, where $A$ is the set of chemical elements, describes the element of all atoms within the unit cells. The atoms' coordinates $X = (x_0, ..., x_N) \in R^{N \times 3}$ describe the coordinates of the atoms within the unit cell. The atoms' coordinates can be Cartesian or fraction coordinates.

**Crystal structure graph representation** Graph structure is commonly used to describe the material crystal structure. Different from molecules, the crystal structure's periodic arrangement and cross-boundary atoms' interactions require a different graph structure. Multi-graph $\mathcal{G} = (\mathcal{V}, \mathcal{E})$ (Xie & Grossman, 2018) is used to represent the material $M$. The node set $\mathcal{V} = \{v_1, ..., v_N\}$ represents the atoms and the edge set $\mathcal{E} = \{e_1, ..., e_N\}$ represents the bonding. While the molecule has a clear definition of bonding, edges in the multi-graph are computed using the k-nearest neighbor with a specific cut-off radius.

### 3.2 CRYSTALLOGRAPHIC SPACE GROUP

The space group of crystal structure consists of a list of symmetry transformations to the atoms within the unit cell. In crystallography, there are 230 space groups (1 to 230). The symmetry level of the crystal structure also increases with the space group number with 1 having the lowest symmetry meaning all atom's positions and lattice parameters are free without following any symmetry operations and constraints.

Each space group has geometrical characteristics in terms of lattice angles or lengths that can be used as constraints to limit the parameters search space. The list of geometrical characteristics is provided in the Supplement Table 5. Given the space group $G_s$, the elements of the space group $g \in G_s$ are a set of symmetry operations. A crystallographic orbit of an atom $o = (x_o, a_o)$ with coordinate $x_o$ and element $a_o$ is defined as

$$O_{G_s}(x_o) = \{g \cdot x_o \mid g \in G_s\} \tag{1}$$

where $g \cdot x_o$ denotes the application of the symmetry operation $g$ on the atom $o$ within the unit cell. From a reference atom $o$, we can obtain a set of equivalent points $O_{G_s}$.

### 3.3 GENERATIVE FLOW NETWORK

The Generative Flow Network (GFlownet) framework is a generative framework that can sample a diverse candidate set proportional to the target reward function. The framework has been successfully applied to many fields such as molecular discovery (Bengio et al., 2021), protein sequence discovery (Jain et al., 2022), causality (Deleu et al., 2022), and continuous control (Li et al., 2023). GFlownet models the sampling process of the compositional object $s$ as the directed acyclic graph (DAG) $G = (S, A)$ where $S$ is the set of states and $A$ is the state transition which is the subset of $S \times S$. The sampling process starts with the initial state vertex $s_0 \in S$ with no incoming edge. and stops at the sink state vertex $s_n \in S$, $n$ is the sampling trajectory length, with no outgoing edge. GFlowNet learns a policy function $\pi$ that can sample the object $x$ with the probability proportional to the non-negative reward function. GFlowNets constructs the object step by step, from the initial state $s_0$ to the sink state $s_n$, forming a trajectory $\tau = s_0, .., s_n, \tau \in \tau$ where $\tau$ is the trajectory set. For any state $s$ in the trajectory, we can define the flow of the state as $F(s) = \sum_{\tau \ni s} F(\tau)$ and the flow of the edge $s \rightarrow s'$ as $F(s) = \sum_{\tau \ni s \rightarrow s'} F(\tau)$ (Bengio et al., 2021). The forward policy that maps the probability of transition from the current state $s$ to the next state $s'$ is given as $P_F(s'|s) = \frac{F(s \rightarrow s')}{F(s)}$. The backward policy mapping probability transition to the previous state $s$ given the current state $s'$ is $P_B(s|s') = \frac{F(s \rightarrow s')}{F(s')}$. The training objective of the GFlowNets is flow matching consistency where the incoming flow is equal to the outgoing flow, $\sum_{s'' \rightarrow s} F(s'' \rightarrow s) = F(s) = \sum_{s \rightarrow s'} F(s \rightarrow s')$, for all the state $s$. Malkin et al. (2022) propose trajectory balance to deal with the long trajectory credit assignment problem. Given the trajectory $\tau = (s_0 \rightarrow s_1 \rightarrow ... \rightarrow s_n)$, the flow of the trajectory is defined as $\prod_{t=1}^{n} P(s'|s)$. The trajectory balance constraint is defined as:

$$Z \prod_{t=1}^{n} P_F(s_t|s_{t-1}) = F(x) \prod_{t=1}^{n} P_B(s_{t-1}|s_t) \tag{2}$$

where $P(s_n = x) = \frac{F(x)}{Z}$ Then the trajectory balance objective is defined as:

$$\mathcal{L}_{TB}(\tau) = \left( log \frac{Z_\theta \prod_{t=1}^{n} P_F(s_t|s_{t-1}; \theta)}{R(x) \prod_{t=1}^{n} P_B(s_{t-1}|s_t; \theta)} \right)^2 \tag{3}$$

## 4 PROPOSED METHOD

Exploring the material space containing hundreds of atoms, while relying solely on terminal rewards, is challenged by the extended planning horizon and the vast combinatorial complexity of the search space. In addition, stable materials only exist in the low-dimension subspace of the material space. Therefore, it is critical to find a new way to systematically model the large state space, state representation, hierarchical policy network, and a physics-embedded reward to improve the exploration.

### 4.1 MATERIAL HIERARCHICAL STRUCTURE STATE SPACE.

We now describe our hierarchical structure state space. We define a hierarchical state space in which a high-level state describes a more abstract concept while the low-level state describes a more discrete concept. The high-level state imposes constraints on lower-level states.

The current design of the state space of crystal structures consists of the lattice parameters space $L$, the atom's elements space $A$, and the atom's 3D coordinate space $X$. This crystal structure state space is flexible, allowing the generative model to freely explore any configuration to find the optimal configurations for the target properties, but it is extremely hard to navigate. A crystal unit cell can have up to hundreds of atoms. Exploring both elements and 3D coordinate spaces for each atom while considering the lattice parameters is extremely difficult due to combinatorial explosion. The crystal structure is highly symmetrical. As a result, atoms' position is only fixed in certain positions ruled by the symmetry operation.

We propose a new crystal structure state space that is more efficient to explore while generating high-symmetry crystal structures with relevant properties. Fig. 1 describes the overall concept of the hierarchical crystal structure state space.

- The symmetry of the crystal structure can be described by the notion of space groups as outlined in Sec. 3.2 that imposes constraints on the lattice parameters, reducing the number of lattice parameters we have to adjust.

- With the list of $n$ symmetry operations provided by the space group, from one atom $o = (x_o, a_o)$, where $x_o$ is the coordinate of the atoms and $a_o$ is the atom's element, we can immediately identify the positions and elements of other $n$ atoms. This effectively reduces the number of searches $n$ times.

- The atoms' interaction within the unit heavily depends on their relative distance from other atoms. The lattice parameters shape the unit cells containing the atoms. Changing the lattice parameters affects the atoms' distance and interaction. Therefore, we place the atom state below the lattice parameters state in our hierarchical state structure.

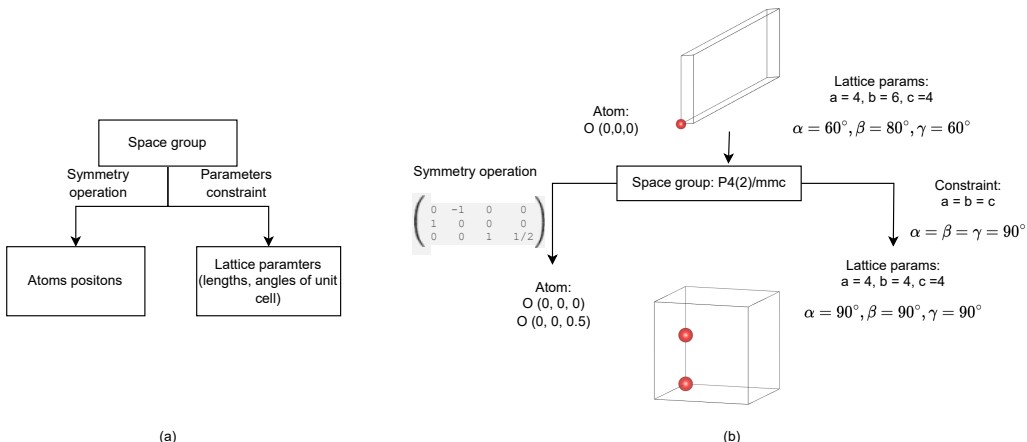

Figure 1: (a) Hierarchical crystal structure state. The space group level provides the set of symmetry operations for atoms' positions and lattice parameters constraints. (b) An example of applying the hierarchical state space. The initial state has one atom O at position (0, 0, 0) with lattice parameters $a = 4, b = 6, c = 4, \alpha = 60°, \beta = 80°, \gamma = 60°$. The space group P4(2)/mmc provides a symmetry operation to generate another atom O at position (0, 0, 0.5). The lattice parameter constraints reduce the unit cell's length search space from $\mathbb{R}^3$ to $\mathbb{R}$ and the unit cell's angles to $90°$

## 4.2 STATE ENCODING.

We now describe the encoding of the state $s_M$ in our CHGFlownet. The state of the material is the state of the unit cell of the material $M = (L, A, X)$. We represent the unit cell of material crystal structure $M = (L, A, X)$ as a multi-graph structure $\mathcal{G}$ as described in Sec.3.1 where $L$ is the lattice parameters, $A$ is the atoms' elements, and $X$ is the atoms' coordinates. We construct the directed multi-graph $\mathcal{G} = (\mathcal{V}, \mathcal{E})$ where $\mathcal{V}$ is the node feature matrix and $\mathcal{E}$ is the adjacency matrix. The node feature is the atom's atomic number and its fraction coordinate within the unit cell.

The graph representation is learned using the Graph Convolution Neural Networks (Kipf & Welling, 2017) with skip connection.

$$\mathcal{H}_\mathcal{G} = GCN(\mathcal{V}, \mathcal{E}) \tag{4}$$

We use the residual skip connection to avoid the vanishing gradient when stacking multiple layers:

$$H^1 = \mathcal{V}, \tag{5}$$

$$F^l(H^{l-1}) = W^{l-1}\sigma(GCN(\mathcal{H}^{l-1}, \mathcal{E})), \tag{6}$$

$$H^l = \sigma(H^{l-1} + F^l(H^{l-1})), \tag{7}$$

where $W$ is the learnable weight matrix, l is the layer's index, and $\sigma$ is a non-linear ReLU activation function. Given the node features of the state graph after GCN layers as $\mathcal{V}'$, the graph-level

representation is obtained using max pooling operation:

$$v'_{\mathcal{G}} = MaxPool(\mathcal{V}') \tag{8}$$
$$h_{\mathcal{G}} = (W_0 + b_0)W_1 + b_1 \tag{9}$$

Lattice parameters are lost when constructing the multi-graph structure. These parameters are important for the policy network to decide when adjusting the lattice parameters. Therefore, we encode additional lattice parameters of the current state as:

$$h_{\mathcal{L}} = MLP([l_1, l_2, l_3, sin(\alpha), cos(\alpha), sin(\beta), cos(\beta), sin(\gamma), cos(\gamma)]) \tag{10}$$

where $l_1, l_2, l_3$ are the lattice lengths, $\alpha, \beta, \gamma$ are the angle of the lattice angle, and $MLP$ is mulilayer perceptron.

It is difficult for the policy network to learn both crystal structure graph representation and decision-making at the same time. We take advantage of the pre-trained model trained on the energy prediction tasks. Specifically, we use the latent vector $H_E$ extracted at the last layer of the Matformer model trained on the formation energy prediction task.

Finally, the crystal structure state $s_M$ is the concatenation of the graph representation $h_{\mathcal{G}}$, the lattice embedding $h_L$, and the pre-trained structure representation $h_E$:

$$s_M = [h_{\mathcal{G}}; h_{\mathcal{L}}; h_E] \tag{11}$$

## 4.3 HIERARCHICAL POLICY

With the hierarchical state space in mind, we propose the hierarchical policy network and flow decomposition. The hierarchical policy consists of two levels, the high-level decision-making policy and the low-level execution policy (Fig. 2). Given the hierarchical crystal state, we define the high-level policy as the space group policy and the low-level execution policy as the atom-lattice policy. The space group policy will handle choosing the space group and apply the space group's constraints on the atom-lattice policy actions.

Due to the hierarchical nature of the state space, our state space is decomposed as $s = (s_{sg}, s_{al})$ where $s_{sg}$ is the space group state and $s_{al}$ is the atoms' and lattice's state. The atoms' and lattice's state consists of lattice parameters $s_{lp}$, atoms' coordinate $s_{ac}$, and atoms' type $s_{at}$ states. Then we define the flow transition as:

$$P(s'|s) = P(s'_{sg}, s'_{al}|s_{sg}, s_{al}) \tag{12}$$

$$P(s'_{sg}, s'_{al}|s_{sg}, s_{al}) = P(s'_{al}|s_{sg}, s_{al}, s'_{sg})P(s'_{sg}|s_{sg}, s_{al}) \tag{13}$$

Then given the trajectory $\tau = (s_0 \rightarrow s_1 \rightarrow ... \rightarrow s_n)$, we then have the trajectory balance constraint (Eq. 2) with the decomposed state as:

$$Z \prod_{t=1}^{n} P_F(s^t_{al}|s^{t-1}_{sg}, s^{t-1}_{al}, s^t_{sg})P_F(s^t_{sg}|s^{t-1}_{sg}, s^{t-1}_{al}) \tag{14}$$

$$= F(x) \prod_{t=1}^{n} P_B(s^{t-1}_{al}|s^t_{sg}, s^t_{al}, s^{t-1}_{sg})P_B(s^{t-1}_{sg}|s^t_{sg}, s^t_{al}) \tag{15}$$

The space group forward and backward transition probability $P_F(s^t_{sg}|s^{t-1})$, $P_B(s^{t-1}_{sg}|s^t)$ is parameterized by the multinoulli distribution defined by the logits output of the space group policy network. The lattice parameters state $s_{lp}$ transition probability is parameterized by the Gaussian distribution defined by the mean $\mu$ and variance $\sigma^2$. The atom fraction coordinates state $s_{ac}$ transition probability is parameterized by the Multivariate Gaussian distribution given by the mean $\mu$ and covariance matrix $\Sigma$. The atom type state $s_{at}$ transition probability is parameterized by the multinoulli distribution.

Then the training objective is the trajectory balance training objective (Eq. 3).

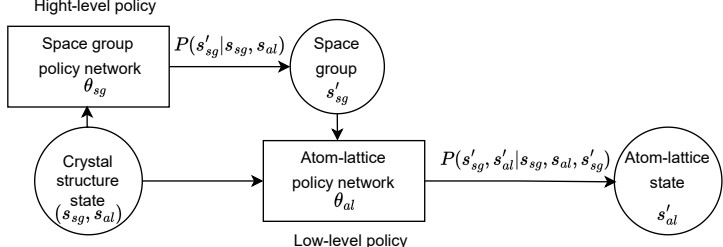

Figure 2: Hierarchical policy for crystal structure state. First, the crystal structure graph state $s$ is decomposed into space group state $s_{sg}$ and atom-lattice state $s_{al}$. Then the transition probability $P(s'_{sg}|s_{sg}, s_{al})$ in Eq. 13 is given by the space group policy network $\theta_{sg}$. The transition probability $P(s'_{al}|s_{sg}, s_{al}, s'_{sg})$ in Eq. 13 is given by the atom-lattice policy network $\theta_{al}$.

### 4.4 PHYSIC-EMBEDDED REWARD FUNCTION

We now describe the physics-embedded reward function. Given the terminal state $s$ at the end of the trajectory, the reward function is the non-negative function $R(x)$ that provides feedback to the model regarding how desirable that terminal state is. We want to generate a valid stable crystal structure. Bringing the physics to the reward function allows the model to learn physics rules directly without having to infer implicitly from the surrogate model.

**Formation energy term** A stable crystal structure requires the formation energy per atom lower than zero. We define the non-negative formation energy term as:

$$R_e(x) = e^{-k(x)} \tag{16}$$

where $k$ is the predicted formation energy per atom given by the prediction model. The energy term encourages the model to generate a crystal structure with low formation energy function.

**Bond distance preferences term** The bond distance preferences term $R_b$ is defined by the distance between any atom pairs $(a_i, a_j)$. The term consists of two validation conditions which are the minimum distance constraint and neighbour distance constraints. The minimum distance constraint term is defined as

$$R_{min}(x) = \begin{cases} \alpha_p, & \text{if } \exists i, j \in x : d(a_i, a_j) < d_{min}(a_i, a_i) \\ 1, & \text{otherwise.} \end{cases} \tag{17}$$

where $d(a_i, a_j)$ is the distance between two $a_i$ and $a_j$ atoms, $d_{(}min)(a_i, a_i)$ is the minimum distance retrieved from the database, and $\alpha_p < 1$ is the penalty term hyper-parameter.

The neighbor distance term is defined as:

$$R_{max} = \begin{cases} \beta_p, & \text{if } \exists j \in x, \forall nei(j), d(a_j, nei(a_i)) > d_{max}(a_j, nei(a_i)) \\ 1, & \text{otherwise.} \end{cases} \tag{18}$$

where $d(a_i, a_j)$ is the distance between two $a_i$ and $a_j$ atoms, $nei(a)$ is the neighbours atoms, $d_{(}max)(a_i, a_i)$ is the minimum distance retrieved from the database, and $\beta_p < 1$ is the penalty term hyper-parameter.

The bond distance preference is defined as:

$$R_{bond}(x) = \begin{cases} \gamma_p, & \text{if } R_{max}(x) < 1 \text{ and } R_{min}(x) < 1 \\ \alpha_p & \text{if } R_{min}(x) < 1 \\ \beta_p & \text{if } R_{max}(x) < 1 \\ 1, & \text{otherwise.} \end{cases} \tag{19}$$

**Density term** Due to the nature of bond distance preference term with strict penalty term for structure violating the minimum distance constraint, the generative model tends to generate a structure

with long distance between atoms pair with only a few neighbor atoms. This leads the model to generate gas with low density rather than material density. On the other hand, structures with very high density (i.e. larger than 10) are unlikely to be realistic as the units are crowded with atoms and formation energy is very high. We measure the generated structure density $P(x)$ and define the density term as the Gaussian function of structure density:

$$R_P(x) = ae^{\frac{-(P(x)-b)^2}{2c^2}} \tag{20}$$

**Composition valid term** To encourage the model to generate the valid composition, we define the composition valid term as:

$$R_{comp}(x) = \begin{cases} 1 & \text{if composition is valid} \\ 0, & \text{otherwise.} \end{cases} \tag{21}$$

**Physic-embedded reward** Given the state $s$, our physics-embedded reward function is defined as:

$$R(x) = (R_e(x) + R_P(x)) * R_{bond}(x) + R_{comp}(x) \tag{22}$$

## 5 EXPERIMENTS

**Battery material discovery task** Motivated by the search for light-weight, transition-metal free cation battery materials, we explore the space of possible materials that can be made from the light elements Be, B, C, N, O, Si, P, S and Cl, and one of the three alkali metals Li, Na and K. This space of materials constitutes materials that can be utilised in lithium-ion, sodium-ion, and potassium-ion battery materials, respectively. Of particular interest to our work is the generation of new transition-metal free solid-state electrolyte materials that can be incorporated in solid-state lithium, sodium, and potassium batteries.

**Baseline** In this section, we describe the baseline methods and implementation details of baseline methods. We compare our proposed generation model CHGFlownet with Generative Flow Networks (GFlownet), Proximal Policy Optimization (PPO) (Schulman et al.), and PGCGM (Zhao et al., 2023). The first baseline method is the GFlowNets with continuous state space. Since the original GFlownNets only works on the discrete space, we adapt the recent works on the continuous space to model the atoms' coordinates and lattice parameters (Lahlou et al., 2023). The second baseline is the PPO with the continuous action space of choosing the atom's fraction coordinate and lattice parameters. The third baseline is PGCGM which learns to sample crystal structure using GAN. PGCGM uses a physics-informed loss function in terms of atoms' distance. To make it fair, in all methods, we only evaluate the generated structure without any further refinement or post-processing.

Regarding the environment used in the experiments, to evaluate the advantage of the hierarchical policy network design and make a fair comparison, both GFlownet and PPO will use the spacegroup-lattice-atoms hierarchical states space proposed in Sec. 4.1. The difference between the proposed method CHGFlownet and baseline methods GFlowNets and PPO is both GFlowNets and PPO baselines have a single-level policy network that outputs space group, lattice parameters, atoms' coordinates, and atoms' type while CHGFlownet follows the hierarchical policy proposed in Sec. 4.3.

### 5.1 MATERIAL VALIDITY

**Validity** We evaluate the proposed method and baseline methods on the validity of the generated crystal structure, measured based on three criteria. We follow previous work (Zhao et al., 2023) for validity measurements a) *CIFs validity* is the percentage of Crystallographic Information Files (CIFs) of generated crystal structure that is readable by pymatgen (Ong et al., 2013) b) *structure validity*: a structure is valid as long as the minimum distance between any two atoms is more than $0.5\mathring{A}$ c) *composition validity*: a composition is valid if the overall charge computed by SMACT Davies et al. (2019) is neutral. From the results in Tab. 1, CIFs validity is an easy condition, and all three methods can achieve a validation rate close to one. Both GFlownet and CHGFlownet structure and composition are nearly identical while PPO has lower structure validity. Without any refinement, PGCGM faces the common problem of generating by sampling methods which is low structure validity. The structure validity and composition validity are used in the reward function described in Sec. 4.4.

Table 1: Validity of the generated structures. We evaluate the top 1000 crystal structures ranked by the reward function after $10^5$ states visited.

| Method | Validity | | |
|---|---|---|---|
| | CIF ↑ | Structure ↑ | Composition ↑ |
| PGCGM | 1 | 0.101 | 0.747 |
| PPO | 0.99 | 0.866 | 0.982 |
| GFlownet | 1 | 0.995 | 0.998 |
| CHGFlownet | 1 | 0.998 | 0.986 |

## 5.2 MATERIAL DIVERSITY AND FORMATION ENERGY

Table 2: Diversity and average of formation energy of the generated structures. We evaluate the top 1000 crystal structures ranked by the reward function after $10^5$ states visited.

| Method | Diversity | | | Formation Energy ↓ | % stable ↑ |
|---|---|---|---|---|---|
| | Crystal family ↑ | Composition ↑ | Structure ↑ | | |
| PGCGM | 2.467 | 2141.196 | 0.524 | 4.558 | 62.5 |
| PPO | 2.549 | 2154.364 | 0.578 | 0.126 | 98.4 |
| GFlownet | 2.544 | 3205.639 | 0.651 | 1.433 | 73.4 |
| CHGFlownet | 2.616 | 3037.808 | 0.761 | 0.882 | 73.4 |

**Diversity** Following method in the previous work (Xie et al., 2022; Zhao et al., 2023), we evaluate both structure and composition diversity of the generated crystal structure. The *structure diversity* is defined as the average pairwise Euclidean distance between the structure fingerprint of any two generated materials (Pan et al., 2021). The *composition diversity* is defined as the average pairwise distance between the composition fingerprints of any two generated materials (Pan et al., 2021). We further use the crystal family defined in Supplement Tab. 5 which is a group of space groups sharing some special geometric characteristics. The *crystal family diversity* is defined as the Shannon–Wiener index (Shannon, 1948) of the number of generated structures in each crystal family.

We evaluate the diversity, average formation energy structures, and the percentage of stable structures discovered by the models (Tab. 2). A stable structure is defined as a structure with formation energy per atom smaller than 2 eV/atom. While GFlownet can find modes with more structure diversity, it has lower composition diversity and average formation energy. As also discussed in previous work (Moksh Jain et al., 2023), reinforcement learning-based method such as PPO is able to discover modes with lower formation energy at the cost of less diversity. Since PGCGM learns from specific crystal distribution and samples from it, the diversity of generated structures is low. Our proposed method has a ***balance*** between diversity and formation energy. The results show that our proposed method CHGFlownet can find the balance in generating both high diversity as well as lower formation energy. In terms of crystal family diversity, as expected, our proposed method is able to generate more diverse crystal families by exploring the space group space.

**Material space discovery** To evaluate the speed of exploring the material space and finding the valid material structure, we evaluate the number of modes found by the generated crystal structure plotted against the number of states visited (Fig. 3). We define a unique *mode* as a valid crystal structure satisfying the distance constraints with negative formation energy. In both environments with step $t = 3$ and $t = 5$, the results show that using our model improves the speed of the mode discovery. Increasing the number of steps indeed makes it harder to discover a mode as there are more atoms to place within the unit cell. With more atoms, finding structures with negative formation energy while maintaining the distance constraint and composition validity is significantly harder. However, on the inherently harder setting, our proposed method still overperforms the other two baselines.

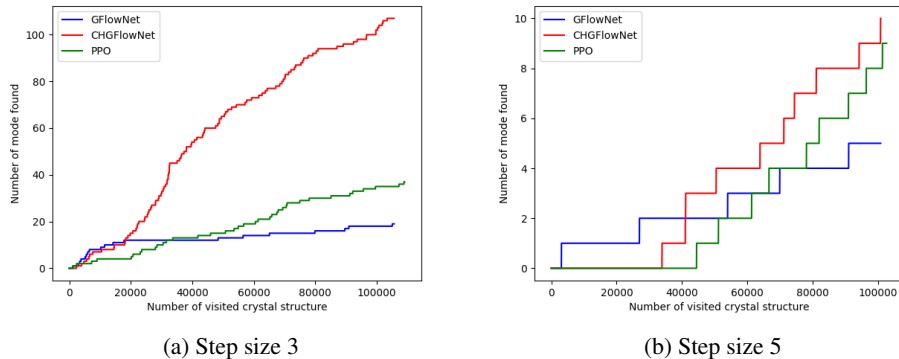

(a) Step size 3               (b) Step size 5

Figure 3: Comparison of CHGFlownet and baselines in crystal structure generation task with the increasing number of choosing elements step corresponding to each column (left: 3 steps, right: 5 steps). The plot presents the number of modes found. A mode is defined as a valid crystal structure with negative formation energy. A step is an action of choosing one atom in the spacegroup-lattice-atom hierarchical state space.

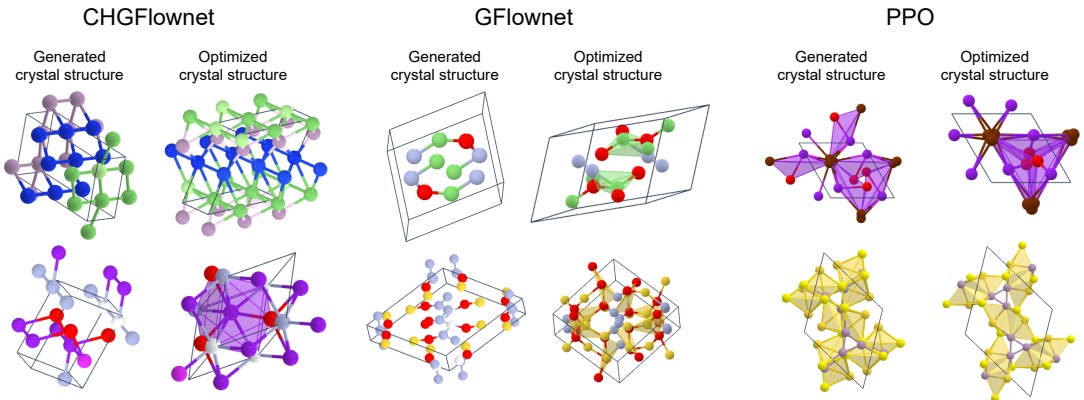

Figure 4: Examples of generated crystal structures and the corresponding structure optimized by M3GNet framework (Chen & Ong, 2022).

## 5.3 STABILITY OF GENERATED MATERIALS

**Crystal structure relaxation** It's common practice that generated crystal structures are then relaxed to lower the potential energy surface using DFT calculation iteratively. However, DFT calculation is expensive. Therefore, it is desirable to generate a structure that is close to energy minima. In this experiment, we evaluate the generated sampled crystal structure compared with the optimized crystal structure. The generated crystal structures are optimized using the M3GNet framework (Chen & Ong, 2022) to iteratively search lower energy surface to lower the energy surface predicted by the potential surface energy model. Examples of generated structures and their corresponding optimized structures are shown in Fig. 4.

**Match rate** We follow the previous work in evaluating the match rate of crystal structure relaxation (Zhao et al., 2023). Given the generated crystal structure $m$ and the corresponding optimized crystal structure $m'$, $m$ and $m'$ are matched if their atoms' translation and angle are within certain thresholds. If the generated structure and optimized structure are matched, we can say that the generated structure is close to optimized, thus more stable. The match rate is the fraction of the number of matched structures on the total number of generated structures.

The result (Tab. 3) shows that our proposed method can produce structures that are nearly optimal in terms of total energy compared to PPO, GFlownet, and PGCGM baseline methods.

Table 3: Match rates of the generated crystal paired with structures optimized by M3GNet (Chen & Ong, 2022)

| Methods | Match rate ↑ |
| --- | --- |
| PGCGM | 0.625 |
| PPO | 0.642 |
| GFlownet | 0.678 |
| CHGFlownet | 0.753 |

## 5.4 ABLATION STUDY

To demonstrate the ability to guide the generative model to generate more stable structures, we perform the ablation study on the reward functions term. We use our CHGFlownet with different reward functions' terms. The generated crystal structures are relaxed using M3GNet optimization framework (Chen & Ong, 2022).

Table 4: The ablation study on the impact of reward function terms on the crystal structure stability. Match rates of the generated crystal paired with structures optimized by M3GNet (Chen & Ong, 2022)

| Methods | Match rate ↑ |
| --- | --- |
| All terms | 0.753 |
| W.o. density | 0.617 |
| W.o. bond score | 0.739 |
| W.o formation energy | 0.734 |

The results (Fig. 4) show that both the density term, bond score, and formation energy term are necessary for the model to generate more stable structures. During the relaxation process, both atoms' positions and lattice parameters are adjusted to lower the total energy and the force of the crystal structure. As a result, the density and the distance between atoms are changed significantly. By using the preference density and distance distilled from prior knowledge such as the relaxed crystal structure dataset, the generation model is able to place atoms and adjust the lattice parameters to maintain the preference distances between atoms.

## 6 CONCLUSION AND FUTURE WORK

We propose a Hierarchical Generative Flow Network for the crystal structure generation task and validate that our proposed method outperforms previous work in terms of sample efficiency material space exploration and stability of the generated crystal structure. In the material discovery field, we believe our method allows domain experts to apply physics and chemistry knowledge to the generation process. In addition, we can construct complex materials with different requirements such as the number of atoms and element requirements thanks to the flexibility of the model. In terms of methodology, since hierarchical state space is a general concept, our framework can easily extend to more complex tasks such as continuous control and robotics where multiple layers of abstraction and discreet states and actions are required.

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

## A  APPENDIX

### A.1  GEOMETRICAL CHARACTERISTICS

Table 5: Geometrical characteristics of space groups in terms of lattice angles or lengths

| Space group | Crystal family | Lengths constraints | Angles constraints | Parameters search space |
|---|---|---|---|---|
| 1-2 | Triclinic c | None | None | $a, b, c, \alpha, \beta, \gamma$ |
| 3-15 | Monoclinic | None | $\alpha = \beta = 90°$ | $a, b, c, \gamma$ |
| 16-74 | Orthorhombic | None | $\alpha = \beta = \gamma = 90°$ | a, b, c |
| 75-142 | Tetragonal | $a = b$ | $\alpha = \beta = \gamma = 90°$ | a, c |
| 143-194 | Hexagonal | $a = b$ | $\alpha = \beta = 90°, \gamma = 120°$ | a, c |
| 195-230 | Cubic | $a = b = c$ | $\alpha = \beta = \gamma = 90°$ | a |

### A.2  IMPLEMENTATION DETAILS

#### A.2.1  STATE GRAPH CONSTRUCTION

We determine the edges of the crystal structure graph using k-nearest neighbor atoms within $4\mathring{A}$. The node feature is the coordinate and the atomic number of the atom.

### A.2.2 FORMATION ENERGY PREDICTION

We use M3GNet (Chen & Ong, 2022) to predict the formation energy. As the M3GNet is only trained on the Material Project valid crystal structure, the predictions for invalid structures may be inaccurate and have abnormally low formation energy. Therefore, we put the negative cut-off for the prediction. Any prediction lower than the cut-off is set to 10 eV/atom.

### A.2.3 BOND PREFERENCE CURATION

## A.3 HYPE-PARAMETERS

Table 6: Hyper-paramters

| Hyper-paramters | Value |
|---|---|
| Learning rate | 0.0001 |
| Learning rate Z | 0.1 |
| Optimizer | Adam |
| Learning rate scheduler $\gamma$ | 1.0 |
| Initial logZ | 0.0 |
| Batch size | 32 |
| $\alpha_p$ (Eq. 17) | 0.1 |
| $\beta_p$ (Eq. 18) | 0.01 |
| $\gamma_p$ (Eq. 19) | 0.001 |