# OpenReview forum: "Hierarchical GFlownet for Crystal Structure Generation"
_ICLR.cc/2024/Conference — Submitted to ICLR 2024_

### Official Review · Reviewer_a89B · 2023-10-30

**Soundness:** 2 fair
**Presentation:** 2 fair
**Contribution:** 2 fair
**Rating:** 3
**Confidence:** 4

**Summary:**

This paper presents a hierarchical GFlowNet designed to generate crystal material structures. The proposed method decomposes the material space into a hierarchy of subspaces, including space groups, lattice parameters, and atoms. The core generative model is the GFlowNet, which is trained using a set of physics-informed reward functions. Experiments have been conducted focusing on battery material discovery to demonstrate the effectiveness of the proposed method.

**Strengths:**

+ The proposed method is straightforward and easy to understand.

+ Enough level of detail on the method is given, easy for reimplementation.

**Weaknesses:**

- The paper lacks clarity in explaining the motivation behind the proposed method and its components, making it challenging to discern the specific advantages or necessity of employing GFlowNet in crystal material generation. The introduction of physics-informed reward functions also lacks motivation. There are totally four new reward functions introduced. While their individual effectiveness is showcased through an ablation study, without a robust theoretical underpinning or discussion compared with existing approaches, it is not convincing enough. Does any existing work try similar stuff already? I could possibly raise some random ideas of some physic-related reward functions but how to prove the authors' proposed set of functions are optimal?

- The writing of the paper feels hurried and overloaded with technical details, at the expense of coherence and contextual depth.  The introduction is full of many technical details but lacks many discussions related to current literature, what the exact problem is which existing methods either do not aim to tackle or cannot deal with. There is no illustrative figure until page 4., hindering a clear comprehension of the proposed components. A high-level overview and an initial presentation of the experimental evaluations and aims could mitigate these issues and enhance the paper's overall accessibility.

- The paper’s experimental section does not convincingly demonstrate the superiority of the proposed method. Observing Tables 1 and 2, it’s evident that the proposed methodology struggles to outperform baseline models consistently across various evaluation metrics. This modest performance raises questions about the fundamental contributions and practical viability of the proposed approach. A clearer demonstration of the proposed method's unique advantages, or a clearer justification of its design choices, is crucial. Without such improvements, the paper’s contribution remains uncertain and inadequately supported.

**Questions:**

- Section 3.1, "atom list L" should be "atom list A"

- Section 3.1, what are the 230 space groups? Add references or explanation or hyperlink to appendix.

- Figure 1 caption is missing period.

- Section 4.1, what are the edges of the graph?

---

> ### Author Response · Authors · 2023-11-23
>
> W1: The paper lacks clarity in explaining the motivation behind the proposed method and its components, making it challenging to discern the specific advantages or necessity of employing GFlowNet in crystal material generation. The introduction of physics-informed reward functions also lacks motivation. There are totally four new reward functions introduced. While their individual effectiveness is showcased through an ablation study, without a robust theoretical underpinning or discussion compared with existing approaches, it is not convincing enough. Does any existing work try similar stuff already? I could possibly raise some random ideas of some physic-related reward functions but how to prove the authors' proposed set of functions are optimal?
>
> A1: On the physical motivation of the minimum bonds information: it was obtained by querying the entire Materials Project database, and computing a bond dictionary for each structure using the function $get\_bonds()$ defined in the OgStructure (https://github.com/sheriftawfikabbas/oganesson/blob/main/oganesson/ogstructure.py). The bond distance between two atoms can fall in a range, but will not go below a specific value for any given structure, otherwise the atoms will experience a repulsive force. The large variability of materials in the Materials Project database is a basis for using a statistical minimum of available bond data for two atoms as a meaningful distance between those two atoms. Such minimum was also used in Ref. [https://www.nature.com/articles/s41524-023-00987-9] for a similar purpose.
> On the physical motivation of the density parameter, we have already explained in the manuscript the reason for considering the density of the generated structure. Quoting our text from the manuscript:
> “Because the bond distance preference term applies a strict penalty for structures violating the minimum distance constraint, the generative model tends to generate a structure with long distances between atoms pairs with only a few neighbor atoms. This leads the model to generate gas with low density rather than solid-state density. On the other hand, structures with very high density (i.e. larger than 10) are unlikely to be realistic as the units are crowded with atoms causing a very high formation energy.”
>
> W2: The writing of the paper feels hurried and overloaded with technical details, at the expense of coherence and contextual depth. The introduction is full of many technical details but lacks many discussions related to current literature, what the exact problem is which existing methods either do not aim to tackle or cannot deal with. There is no illustrative figure until page 4., hindering a clear comprehension of the proposed components. A high-level overview and an initial presentation of the experimental evaluations and aims could mitigate these issues and enhance the paper's overall accessibility.
>
> A2: As discussed in the introduction, in this work, we aim to tackle the problem of the complexity associated with the large search space (in this case is the large material space) by proposing a new generative model termed Hierarchical Generative Flow Networks.
>
> The paper’s experimental section does not convincingly demonstrate the superiority of the proposed method. Observing Tables 1 and 2, it’s evident that the proposed methodology struggles to outperform baseline models consistently across various evaluation metrics. This modest performance raises questions about the fundamental contributions and practical viability of the proposed approach. A clearer demonstration of the proposed method's unique advantages, or a clearer justification of its design choices, is crucial. Without such improvements, the paper’s contribution remains uncertain and inadequately supported.
>
> W3: The method's unique advantage is the ability to incorporate constraints into the sampling process and soft constraint to rewards function.

---

### Official Review · Reviewer_SEfh · 2023-10-31

**Soundness:** 3 good
**Presentation:** 2 fair
**Contribution:** 2 fair
**Rating:** 5
**Confidence:** 4

**Summary:**

In this paper, the authors propose CHGFlowNet, a novel application of GFlowNet for crystal generation. The proposed framework employs a hierarchical approach to sample the crystal structure at two levels: the high-level policy determines the space group symmetry of the entire crystal, while the low-level policy refines the lattice parameters and atomic positions based on the space group constraints. The reward functions of CHGFlowNet consider the formation energy, bond distances, structure density and composition validity. Experiments demonstrate the effectiveness of the hierarchical modeling and the incorporation of physics priors.

**Strengths:**

1. The paper focuses on crystal generation, a significant problem in materials science, and is the first to apply GFlowNet to address this challenge.

2. The authors utilize a bilevel generation framework to capture space group symmetry and combine four types of physics-inspired reward functions as priors. The experiments provide evidence for the effectiveness of these approaches.

**Weaknesses:**

1. The presentation of the proposed method lacks important details. For instance, the paper does not mention the initialization method. To improve understanding, it would be beneficial to provide a detailed algorithm outlining the generation process.

2. The proposed method does not consider translation invariance. In Section 4.1, fractional coordinates are directly used as inputs for the GCN model, which violates the translation invariance of 3D crystals. This issue has been well-studied in previous predictive [1] and generative [2] methods, and its omission is a notable weakness.

3. The paper does not include comparisons with some relevant baseline methods. Specifically, the authors do not compare their approach with CDVAE [2], a method focused on general crystal generation that could be applicable to the tasks presented in this paper. Additionally, the paper lacks a thorough discussion of recent generative methods in the field [3,4]. Addressing these omissions would strengthen the evaluation of the proposed method.

**Questions:**

1. As mentioned in Weakness 1, how to sample the initial state?

2. The paper does not discuss Wyckoff positions [5], which are crucial for understanding how the proposed method handles space groups with large multiplicities. Consider a space group with n group elements (i.e. symmetry operations), not all atoms are copied n times. Some of the "replicas" are overlapped, leading to special Wyckoff positions. For instance, space group Fm-3m (No. 225) has a maximum multiplicity of 192. How does the proposed method manage a system with such a large multiplicity? Are atoms duplicated 192 times, or is there a deduplication strategy to determine the special Wyckoff positions?

3. (Minor) In Section 4.3, the authors propose four reward functions, but only three are analyzed in the ablation studies. Could the authors provide insights into the impact of the composition validity term on the overall performance of the proposed method?

[1] Yan, Keqiang, et al. "Periodic graph transformers for crystal material property prediction." Advances in Neural Information Processing Systems. 2022.
[2] Xie, Tian, et al. "Crystal Diffusion Variational Autoencoder for Periodic Material Generation." International Conference on Learning Representations. 2021.
[3] Jiao, Rui, et al. "Crystal Structure Prediction by Joint Equivariant Diffusion." arXiv preprint.
[4] Luo, Youzhi, et al. "Towards Symmetry-Aware Generation of Periodic Materials." arXiv preprint.
[5] https://en.wikipedia.org/wiki/Wyckoff_positions

---

> ### Author Response · Authors · 2023-11-23
> **Regarding the Weaknesses of the paper**
>
> W1: The presentation of the proposed method lacks important details. For instance, the paper does not mention the initialization method. To improve understanding, it would be beneficial to provide a detailed algorithm outlining the generation process.
>
> A1:  We provided the initialization as well as the sampling process in Algorithm 1.
>
> W2: "The proposed method does not consider translation invariance. In Section 4.1, fractional coordinates are directly used as inputs for the GCN model, which violates the translation invariance of 3D crystals. This issue has been well-studied in previous predictive [1] and generative [2] methods, and its omission is a notable weakness.
>
> A2: The atomss list only mantain a reference atom of crystallographic orbit to make sampling process light-weight. Because it is not a complete crystal structure, translation invariance is not required. At the end of the trajectory where the complete crystal structure is generated to get the formation energy, the graph representation as well as the formation energy model is SE(3) invariant.

---

### Official Review · Reviewer_45US · 2023-10-31

**Soundness:** 1 poor
**Presentation:** 1 poor
**Contribution:** 2 fair
**Rating:** 3
**Confidence:** 5

**Summary:**

This paper tackles the problem of generating crystal structures with generative models. In particular, this paper uses the recently introduced generative flow networks (GFlowNets), which allow for the generation of diverse objects in a compositional space. This paper leverages some of the advantages of GFlowNets to design a state space that decomposes crystals into their space group, lattice parameters and atom positions, by preserving some of the geometrical properties imposed by the sequential selection of these crystal features. The authors propose to introduce further constraints via the reward function of the GFlowNet, which consists of a combination of multiple rewards that introduce a penalty if a condition is not met. The evaluation procedure studies the validity, diversity and formation energy of generated crystals, compared to the Physics Guided Crystal Generative Model (PGCGM) and a simpler GFlowNet without some of the aforementioned features.

**Strengths:**

There are some strong aspects of this paper I would like to highlight.

First of all, the authors have identified some of the current challenges in the problem of crystal structure generation that can be potentially tackled by the relatively recent framework of GFlowNets. In particular, the abstract already mentions the difficulty of discovering new crystals due to the vast search space and the rarity of stable materials in this space, as well as the need for diversity in the exploration and discovery process. These are indeed the challenges that GFlowNets have been shown to address effectively. Furthermore, the authors have also leveraged the potential of GFlowNets to generate objects in a compositional space by designing a generation graph that sequentially reduces the search space.

In particular, in my opinion there are good ideas in the design of the generation process, such as the application of the symmetry operations from a previously selected space group to sample atom positions, as well as the constraints imposed by the space group on the lattice parameters. I also positively receive the fact that the authors have proposed an approach to incorporate information regarding the inter-atomic interactions (see my comments in the next section regarding the specific approach proposed).

Another strong point is the concise but insightful revision of the literature on crystal structure generation.

**Weaknesses:**

I also have some major concerns that generally dominate my assessment of the paper. In broad terms, my main concerns have all to do with the rigour of the presentation and claims, as well as the fact that the descriptions of important parts of the presented methods are too vague for me to be able understand the details, despite having first-hand experience with both GFlowNets and crystal structure generation. Below, I will elaborate on some specific examples.

An important aspect that is very vaguely described is the procedure to sample atom positions with GFlowNets. After carefully studying the manuscript, I only found a couple of places that tangentially refer to the procedure to obtain atom positions, which, crucially, is not trivial in the task of crystal structure generation. In Figure 1, a flowchart on the left contains a box with the text "Atom positions", but without any further information. The caption of the figure provides a little more information, but still not enough for me to form a sufficiently informed understanding of the method: "The initial state has one atom O at position (0, 0, 0) [...]. The action of choosing space group P4(2)/mmc provides a symmetry operation to generate another atom O at position (0, 0, 0.5)." This statement refers implicitly to one of the Wyckoff positions of space group P4(2)/mmc (international number 131), with multiplicity 2, but no further information is provided about, for example: why does the initial state has one atom at (0, 0, 0)? How are other atom positions sampled? What is the generation process to sample the total number of atoms in a crystal?

Section 4.2 contains more information about the selection of atom coordinates: "The atom fraction coordinates state $s_{ac}$ transition probability is parameterized by the Multivariate Gaussian distribution given by the mean $\mu$ and covariance matrix $\Sigma$". While this seems to provide direct information about the process to select atom positions, no further details are provided, which leaves me with more questions than answers. First, the questions posed above remain, for example, how many atoms are selected? Assuming a fix number of atoms, are they sampled at once, one by one? Second, the fact that a Gaussian distribution is used to sample coordinates is surprising, since the Gaussian distribution is unbounded and can for instance sample negative (fractional) coordinates, which would be invalid. Alternatively, I suggest using a Beta distribution, as in Lahlou et al. (2023), which seems to better suit this use case.

These questions are relevant not only because atom coordinates in crystal are subject to very specific geometric and physical constraints in a crystal - some of which are addressed in the method - but also because they involve sampling in a continuous state space, which has proven to be notoriously non-trivial, as one has to make sure that the set of assumptions discussed by Lahlou et al. are properly preserved. In other words, a naive design of a hybrid (mix of discrete and continuous) state space is likely to be incorrect. Nonetheless, the only mention to this aspect in the present paper is the following: "As the original GFlowNets only work on the discrete space, we adapt it to work on the continuous space of the atoms’ coordinates and lattice parameters (Lahlou et al., 2023)". In this regard, I would also like to mention that the phrasing may be misleading, as it seems to indicate that the adaptation to continuous spaces is introduced in this paper, despite providing the correct reference.

I have elaborated on the question of the selection of atom coordinates, but it is not the only aspect that is vaguely described. Other aspects are the selection of lattice parameters, the criteria used in the evaluation ("A structure $m$ and its optimized structure $m'$ are matched if their atoms translation and angle are _within certain thresholds_" (emphasis mine), the criteria for diversity in the evaluation, the definition of a mode in Section 5.2, etc.

An essential aspect of all GFlowNets including the particular instance presented in this paper is the reward function. While Section 4.3 describes the decomposition of the reward function into _sub-rewards_ that encode various (soft) constraints, the main component of the reward function is the formation energy of the sampled crystal. The only piece of information provided in the paper is the following: "$E(x)$ is the predicted formation energy per atom given by the prediction model". However, no further mention or reference of the model is provided and the reader is left to guess what that model might be, including throughout the evaluation procedure, which crucially depends on the formation energy itself. After inspection of the supplementary material, one can find a short, non-referenced section (A.2.2) that mentions M3GNet as the predictive model. Incidentally, this section also contains vague descriptions: "Therefore, we put the negative cut-off for the prediction. Any prediction lower than the cut-off is set to 10 eV/atom." What is the cut-off? How often does it occur?

Without getting into details, I would like to mention other parts of the manuscript that either lack details or are potentially incorrect:

- The lattice parameters used to illustrate the sampling process in Figure 1 (a = 4, b = 6, c = 4) do not seem to match the drawing.
- "HGFlowNets generalize GFlowNets": this seems like a strong claim that receives no further attention in the paper. I would have appreciated an elaboration on how the proposed method generalises GFlowNets.
- "The key insight to solving the large state space problem is breaking space exploration into more meaningful hierarchical sub-tasks. Here the higher-level tasks explore more meaningful actions that are closely related to the reward function while lower-level tasks handle the configuration adjustment corresponding to the action taken at higher-level tasks." It is unclear to me what the authors mean by "more meaningful actions", why the are closer to the reward function (what do they mean by "close"?). I understand that the authors are describing the inherent hierarchical decomposition of the generating graph of GFlowNets. If that is the case, I would note that although the authors seem to claim the novel introduction of this notion, this is an intrinsic feature of GFlowNets, which rely on the decomposition of the sampling process to enable generalisation.
- "In the hierarchy, lower states represent discrete concepts constrained by the higher states that represent more abstract concepts" What do the authors mean by "discrete concepts"? It would be great if the authors could provide a more formal definition if possible, or more details of their intuition.
- The paper contains a number of typos or errors in the description of the methods. For example:
    - "a triplet (L, A, X) of lattice parameters L, atom list L, and atom coordinates X"
    - "$\tau = s0, \ldots, s_n, \tau \in \tau$ where $\tau$ is the trajectory set" (note the repetition of lowercase $\tau$ to indicate the trajectory set).
    - "for all the state s"
    - "the flow of the trajectory is defined as..." Here the authors are defining the forward probability of a trajectory, not the flow of a trajectory.
    - To indicate atom position, the authors use both lowercase $o$ and uppercase $O$, inconsistently.
    - In Section 4.2, the authors provide probability of transitions, but mention "flow transitions" in the text.
    - In general, it is hard to easily make sense of the equations in Section 4.2. I wonder if they shed more light than confusion.

While I mentioned that a strength of the paper is the incorporation of certain constraints in the method, I would like to note that the specific approach via additional terms in the reward function is suboptimal to the capabilities of GFlowNets. The compositional nature of GFlowNets allows for the incorporation of hard constraints in the sampling process, while penalties as reward terms are only soft constraints. Further, this approach introduces multiple hyper-parameters (all the coefficients of the rewards) that I also see as a weakness.

Regarding the evaluation, I would like to first note that is also hard to draw insights because of the lacks of details about the training procedure and evaluation criteria (as mentioned above). From a direct analysis of the results provided in the tables, we can see that the method proposed by the authors generally obtains better metrics than a simpler GFlowNet (no details provided either) and that the recently published PGCGM. For example, the average formation energy of the PGCGM, GFlowNet and CHGFlowNet (method introduced in the paper) are 4.558, 1.433 and 0.882, respectively. From a naive interpretation, we would conclude that CHGFlowNet is better because the average formation energy is lower. However, first of all an average positive formation energy of the top samples is practically not useful, since the formation energy should be negative for the materials to be potentially stable, as discussed by the authors of PGCGM, for instance. Second, in the PGCGM paper, the authors report that 39.6 % of the generated structures have negative formation energy. While both results are in principle compatible, the PGCGM paper provides a plot of the distribution of the formation energy, showing a mode at around 0 eV/atom, that is far from the average 4.558 reported in this paper. This example, together with the lack of details cast doubt on the results provided in the paper.

Finally, I would like to mention the existence of concurrent work with significant overlap with the present paper, which I suppose the authors might find relevant: https://arxiv.org/abs/2310.04925

**Questions:**

I have organised most of my questions for the authors in my discussion of the weakness. Therefore, I kindly refer the authors to the previous section regarding my questions.

---

> ### Author Response · Authors · 2023-11-23
> **Regarding the Weaknesses of the paper**
>
> Thank you reviewer for your very detailed comments. It helps us to improve our manuscript alot. We really appreciate your comments regarding our manuscript. We hope we can answer your concerns.
>
> W1: In Figure 1, a flowchart.
>
> A1: Thank you for pointing this out. The wording in the Fig. 1 is indeed misleading. It should be worded "The current state". The figure is to explain how an action of choosing an atom reference position can be translated into multiple positions with symmetry operation.
>
> W2: Sampling process, Gaussian distribution.
>
> A2: Each step in the trajectory samples one single atom coordinates $x_0$.  Then the sampled atom is added to the list of sampled atoms $A_{sampled}$. $A_{sampled}$ only stores the one reference atom $x_0$ of the rystallographic orbit (Eq. 1). At the end of the trajectory, we apply the symmetry operation given by the sampled space group to have the full list of atoms $A_{crystal}$.
>
> During the trajectory sampling, we only use the graph representation from the sampled atoms list $A_{sampled}$ as the policy network input. There are two reasons we chose this implementation design. First, with the graph representation is permutation invariant. Second, if we apply symmetry operation on the sampled atoms list $A_{sampled}$ and use the full crystal structure atom list $A_{crystal}$, the number of atoms of the crystal structure can easily reach to hundred or thousand of atoms due to high multiplicity. The computation of a crystal graph of a thousand atoms is very computationally expensive.  Creating a new crystal graph at every steps of the trajectory is not preferred.
>
> W3: Continuous GFlownet assumptions, negative fraction coordinate
>
> A3: We also change the claim to "As the original GFlowNets only work on the discrete space, we follow the recent work on continuous GFlownet (Lahlou et al., 2023)  to work on the continuous space of the atoms’ coordinates and lattice parameters."
>
> It is true that sampling from Gaussian distribution may result in the negative fraction coordinate. Therefore, we cap the sampled action to $[0,1]$ to ensure the validity of the action.
>
> W4: Other aspects are the selection of lattice parameters, the criteria used in the evaluation ("A structure
>  and its optimized structure
>  are matched if their atoms translation and angle are within certain thresholds" (emphasis mine), the criteria for diversity in the evaluation, the definition of a mode in Section 5.2, etc.
>
> A4: We have provided more information regarding sampling lattice parameters, criteria for stability experiments, and definition of a mode as following and added to Appendix section.
>
> W5: Prediction model
>
> A5: We add the reference of the prediction model in Sec. 4.3 to A.2.2 . We set the cut-off to $-10$ eV/atom as any structure having lower formation energy than that is more likely to be invalid. It is very difficult to tell exactly how often it occurs as the prediction model is a black box function. We can not tell exactly which type of structure causes this. In our own experience, it rarely happens, less than 10 times out of 100000 sampled crystal structures.
>
> W6: The lattice parameters used to illustrate the sampling process in Figure 1 (a = 4, b = 6, c = 4) do not seem to match the drawing.
>
> A6: We have adjusted the Figure to match the number.
>
> W7: "HGFlowNets generalize GFlowNets" claim
>
> A7: We removed this claim as this requires further experiments on different tasks to fully support this claim.
>
> W8: "more meaningful actions" term
>
> A8: The term "more meaningful actions" means that the actions are designed to have a more direct connection the target properties. The connection in the design is provided by prior knowledge, for example, perovskite structure class with conductivity. As discussed in the third paragraph in the Introduction section, the crystal structure class can  associate with the properties that we are optimizing. However, to generate the crystal structure, we need to sample the lattice parameter and the atoms' coordinates. Instead of only relying on very complex interactions between atoms using only coordinates to optimize properties, we can choose simpler actions but can directly associate with target properties. In this paper, we limit our experiment to only space group. In future applications, we can extend to more sophisticated higher-level tasks such as choosing composition to associate with crystal structure neutral charge.

---

> ### Author Response · Authors · 2023-11-23
>
> W9: "discrete concepts" term
>
> A9: The term "discrete concepts" means that the actions or states are directly involved in the sampling process and can not be omitted. For example in the crystal structure generation task, we cannot have a structure without lattice parameters or atoms' type and coordinates. Meanwhile, the action of choosing a spacegroup is optional but can greatly assist the sampling process. Another example is in the control task. The action of moving the robot's joint is necessary and directly linked to the robot's movement. On the other hand, the action "move left", "move right", or "move to the door" is more at an abstract level.
>
> W10: The paper contains a number of typos or errors in the description of the methods.
>
> A10: Thank you for pointing out these errors. We have corrected those accordingly.
>
> W11: Soft constraint is not optimal, hard constraint during the sampling process.
>
> A11: Our proposed method in crystal structure generation incorporate both hard constaints and soft constraint. The hard constraint is the task we performed in the experiments. The generated crystal must have one of three alkali metals Li, Na, and K with coupled with other light elements Be, B, C, N, O, Si, P, S and Cl. To generates the required material, the hard constraint is choosing the one of three alkali and must coupled with other light elements. which is imposed during the atom's element sampling. The soft constraints are imposed in the reward functions.
>
> W12: PGCGM results
>
> A12: PGCGM performs the post-generation step to refine the crystal structure. To make it fair, in all methods, we only evaluate the generated structure without any further refinement or post-processing. As we pointed out in the 1st paragraph of Sec 5.1, PGCGM faces the common problem of sampling from data-induced distribution without any
> refinement, which is low structure validity and atoms are very close to each other thus leading to high formation energy.
>
> W13: concurrent work
>
> A13: Thank you for introducing us to the concurrent work. Our work is different as we sample the complete crystal structure while other work only samples crystal structure parameters without atom coordinates. The formation energy is only provided by a newly trained prediction function without any coordinates.

---

### Official Review · Reviewer_ni6Z · 2023-11-06

**Soundness:** 3 good
**Presentation:** 4 excellent
**Contribution:** 4 excellent
**Rating:** 5
**Confidence:** 4

**Summary:**

The paper presents an extension of GFlowNets designed specifically for the generation of crystal structures. The proposed method Crystal Hierarchical Generative Flow Network  (CHGlownet) utilizes the hierarchical structure for crystals to more efficiently explore the space of possible crystals to find structures with desired properties.

The proposed approach incorporates a hierarchical policy for the crystal structure that is composed of determining the crystal space group on a higher level and the atomic lattice that consists of the crystal lattice, the atom coordinates, and atom types. The method also includes a physics-informed reward function.

The approach is validated on the Battery material discovery task where it presents with improved performance over the PGCGM method and the 'flat GFLowNet.

**Strengths:**

The proposed approach is well-motivated and presented clearly. A good background has been presented on the crystallography task.

The empirical analysis is aligned with the key goals of the material discovery task, assessing diversity and stability as well as an ablation study.

**Weaknesses:**

The experimental section seems quite weak. The method is tested on one task and one dataset. There is only one baseline model and the original GFlownet is used as a baseline.

There are many tasks such as the generation of MOF or Zeolites that are very well suited for the evaluation of this extension of GFlownets. There is a large number of methods that form the state of the art on these tasks that would be great candidates for baselines.

**Questions:**

Given that the relaxation results in such large changes in the lattice structure and the atom positions, how much of a role does the lower level (in the hierarchy) policy actually play a role in the generation process?

Can the crystal be defined only by the atom positions without the lattice parameters? Even though these two data structures are definitely coupled, does the model generally treat them independently?

---

> ### Author Response · Authors · 2023-11-23
> **Regarding the Weaknesses of the paper**
>
> Thank you reviewer for your valuable time and consideration. We really appreciate your comments regarding our manuscript. We hope we can answer your questions.
>
> W1: The experimental section seems quite weak. The method is tested on one task and one dataset. There is only one baseline model and the original GFlownet is used as a baseline.
>
> A1: In the experiment, our goal is to compare the proposed method with the representative methods of different deep learning approaches introduced in the related works section. So in this case, we choose PGCGM for the sampling approach, GFlownet for iterative generation approach.
>
> W2: There are many tasks such as the generation of MOF or Zeolites that are very well suited for the evaluation of this extension of GFlownets. There is a large number of methods that form the state of the art on these tasks that would be great candidates for baselines.
>
> A2: Thank you very much for the task suggestion. Both MOF and Zeolites generation task generates crystal structure based on specific requirements. MOF generates structures with crystalline porous materials with high surface area and tunable pore sizes.  Zeolites mainly consist of silicon, aluminium, oxygen with formula as $M^{n+}_{1/n}(AlO_2)^{-}(SiO_2)_x\cdot y H_2O$. In our experiment section, we demonstrate the flexibility based on our battery material discovery task, generating structure with defined formula $M_x N_y A_z$ where A is one of three akali metals Na, K, and Li, M and N are light elements Be, B, C, N, O, Si, P, S and Cl. All these three tasks can be used interchange to demonstrate the iterative crystal structure generation.
>
> Q1: Given that the relaxation results in such large changes in the lattice structure and the atom positions, how much of a role does the lower level (in the hierarchy) policy actually play a role in the generation process?
>
> A1: Structure relaxation is a common step in crystal structure discovery.  The relaxation process may result in small changes or large changes in the lattice parameters and atoms' position. If the lower level (in the hierarchy) policy can learn to optimize the reward function, then the change to the lattice parameters and atoms' position should be small. Then we can say that the generated structure from the framework is close to optimal. As a result, it will reduce the time run for DFT optimization.
>
> Q2: Can the crystal be defined only by the atom positions without the lattice parameters? Even though these two data structures are definitely coupled, does the model generally treat them independently?
>
> A2: For the lattice and atoms position, we follow the previous works using the lattice parameter and atoms' fraction coordinate to define a crystal structure. When the policy network takes actions of choosing lattice parameters and atoms' positions, it also has to consider the distance between atoms due to the bond distance preferences term in the reward function. Therefore, the policy needs to treat them dependently.

---

### Meta-Review · Area_Chair_XiRC · 2023-12-04

**Metareview:**

This paper introduces a hierarchical generative model for crystal structure generation based on Generative Flow Networks (GFlowNets).

The reviewers appreciate that the studied problem is important, that GFlowNets offer advantages that are a good fit for the problem of crystal structure generation, and that the paper provides a comprehensive background/literature review on the topic.

Overall, however, the paper does not currently meet the bar for acceptance as there were several concerns about insufficient or unclear detail on the proposed method and training procedure in the submitted paper. The reviewers further raised concerns that the experiments do not convincingly validate the claims of the paper.

**Justification For Why Not Higher Score:**

Scientific claims are not sufficiently validated and paper lacks detail on method and training procedure.

**Justification For Why Not Lower Score:**

N/A

---

### Decision · Program_Chairs · 2024-01-16

Reject